# 🌿 AMBROSIA: A Benchmark for Parsing Ambiguous Questions into Database Queries

**Irina Saparina**     **Mirella Lapata**
Institute for Language, Cognition and Computation
School of Informatics, University of Edinburgh
10 Crichton Street, Edinburgh EH8 9AB
i.saparina@sms.ed.ac.uk     mlap@inf.ed.ac.uk

## Abstract

Practical semantic parsers are expected to understand user utterances and map them to executable programs, even when these are ambiguous. We introduce a new benchmark, AMBROSIA, which we hope will inform and inspire the development of text-to-SQL parsers capable of recognizing and interpreting ambiguous requests. Our dataset contains questions showcasing three different types of ambiguity (scope ambiguity, attachment ambiguity, and vagueness), their interpretations, and corresponding SQL queries. In each case, the ambiguity persists even when the database context is provided. This is achieved through a novel approach that involves controlled generation of databases from scratch. We benchmark various LLMs on AMBROSIA, revealing that even the most advanced models struggle to identify and interpret ambiguity in questions.

## 1 Introduction

Semantic parsing translates natural language utterances to logical forms or executable programs in some machine-readable language (e.g., SQL). It has emerged as an important component in many real-world applications (Őzcan et al., 2020; Liang, 2016; Wang et al., 2023b; Dukes, 2014) as it allows users to seek information and control computer systems naturally and flexibly in natural language. Practical semantic parsers are expected to understand user utterances and map them to executable forms, even when these are ambiguous (see Figure 1 where a user request allows multiple interpretations, each corresponding to a different logical form). Ambiguity is a pervasive challenge in natural language applications (Min et al., 2020; Liu et al., 2023a; Yuan et al., 2023), and semantic parsing is no exception. Wang et al. (2023a) show that more than half of failure cases for a text-to-SQL semantic parser are due to ambiguity which can occur in different forms and at different levels.

Although the problem of mapping natural language utterances to formal representations has been studied extensively, the issue of ambiguity has received less attention. Stengel-Eskin et al. (2024) evaluate the ability of large language models to parse ambiguous sentences to first-order logic, focusing on five well-known linguistic ambiguities. In the context of text-to-SQL parsing, other work (Wang et al., 2023a; Bhaskar et al., 2023) introduces vagueness into the questions of popular benchmarks like Spider (Yu et al., 2018) by modifying their databases, e.g., through synonyms. Although targeting a real-world application, database augmentation is limited to a single type of ambiguity[1] and often operates in an artificial setting. For example, consider the database shown in Figure 1b. We could add a "Scriptwriters" table with the same content as the existing "Screenwriters" one. Our new database would allow vague questions, but would not be very realistic or well-designed.

---

[1]Vagueness and ambiguity are often considered distinct properties (Frappier et al., 2012); however, for simplicity, we will refer to vagueness as a type of ambiguity.

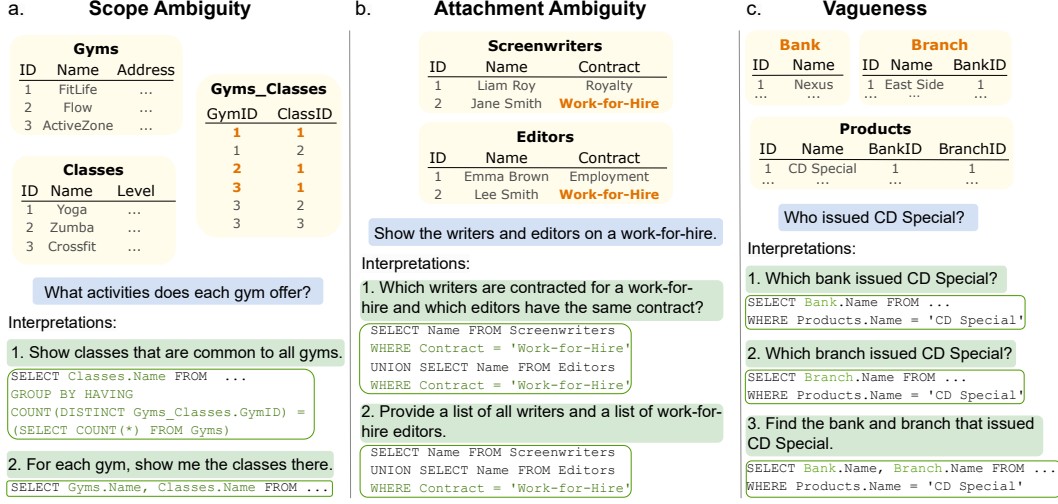

Figure 1: Types of ambiguous questions (highlighted in blue), their interpretations (highlighted in green), and corresponding SQL queries. Database elements that could lead to ambiguity are highlighted in orange.

In this paper we introduce AMBROSIA, a novel benchmark which we hope will both inform and inspire the development of parsers capable of recognizing and interpreting ambiguous queries. AMBROSIA covers 16 distinct domains, it contains 846 multi-table databases, ambiguous questions, their unambiguous interpretations provided by humans, and complex SQL queries (4,242 in total). It includes three types of ambiguity, i.e., scope ambiguity, attachment ambiguity, and vagueness, showcasing a diverse range of SQL queries. Figure 1 shows examples of ambiguous questions in blue blocks and their unambiguous interpretations in green, both of which can be phrased as requests. Aiming to mimic real-world semantic parsing scenarios with realistic and diverse databases, we create them automatically in three steps: (a) we specify a domain of interest (e.g., Banking); (b) we generate key concepts and relations such that they obey constraints imposed by a chosen ambiguity type (e.g., for scope ambiguity, the database must have a many-to-many relationship with a common element; see Figure 1a where multiple gyms offer the same class, namely yoga); and (c) we generate SQL statements to construct tables with the desired structure. We use a large language model for generation (OpenChat; Wang et al. 2024) and view database creation as a semantic parsing problem. Since we can automatically filter predicted SQL statements based on execution results, minimal manual effort is required to validate the generated databases and their content.[2]

We benchmark multiple advanced large language models of different sizes on AMBROSIA, including the most recent Llama 3 (Dubey et al., 2024) and GPT-4o. Our experiments reveal that models struggle to recognize ambiguity and provide all possible SQL queries for all interpretations. They often capture only one interpretation and are biased towards a specific type of ambiguity. The best model, Llama 3-70B, achieves only 31% recall on ambiguous questions compared to 66% on unambiguous ones. AMBROSIA offers a diverse range of questions from various domains, each introducing distinct types of ambiguity along with their interpretations. This diversity offers invaluable insights into the challenges of real-world semantic parsing.

## 2 Related Work

The ambiguity inherent in natural language has been studied through the lens of various tasks, including question-answering (Min et al., 2020), natural language inference (NLI; Liu et al. 2023a), and coreference resolution (Yuan et al., 2023), where models have been broadly found lacking in their ability to resolve ambiguities. The bulk of previous work has focused on question answering, with emphasis on asking clarification questions to understand user intent (Rahmani et al., 2023), open-domain question answering where a query can plausibly have multiple valid answers (Min et al.,

---

[2]The code and data are publicly available at: `ambrosia-benchmark.github.io`

2020), disambiguating database search results in the context of task-oriented dialogue systems (Qian et al., 2022; Kim et al., 2023), and leveraging relevance feedback to rerank the answers returned from a QA engine based on knowledge graphs (Liu et al., 2023b).

Within the broader area of semantic parsing, some work (Li et al., 2023a; Mu et al., 2024) has concentrated on clarifying vague questions for code generation. There is also interest in creating datasets with ambiguous utterances and corresponding logical form representations. Rasmussen and Schuler (2020) collect a dataset of $\lambda$-calculus translations that includes examples of scope ambiguity, Arthur et al. (2015) explore different types of ambiguities that arise in the task of mapping search queries into SCFGs, and Stengel-Eskin et al. (2024) create a benchmark for mapping ambiguous sentences into first-order logic. We also collect ambiguous utterances, however, our benchmark is designed for parsing questions into SQL database queries. Unlike Arthur et al. (2015) and Stengel-Eskin et al. (2024) who create synthetic examples from templates, we ask human annotators to write natural questions for a real-world application.

Related work in text-to-SQL parsing has primarily focused on vague questions. Wang et al. (2023a) detect questions containing ambiguous tokens that could map to multiple columns. Their dataset builds on WikiSQL (Zhong et al., 2017) and Squall (Shi et al., 2020) which are limited to *single-table* databases. Bhaskar et al. (2023) modify Spider (Yu et al., 2018) with ChatGPT to create databases that exclusively support vague questions. Despite relying on Spider, their approach often yields unrealistic databases, e.g., they introduce ambiguity in table names by copying and renaming existing tables, which leads to information being duplicated. Our dataset not only supports vagueness but also includes scope and attachment ambiguities. Moreover, AMBROSIA provides *multi-table* databases that mirror realistic applications.

Huang et al. (2023) explore ambiguity in the KaggleDBQA dataset (Lee et al., 2021) focusing on vagueness, underspecified output formats, and unknown data structures. In contrast, our work assumes the database context is fully specified and focuses on different types of linguistic ambiguity. Other work Floratou et al. (2024); Pourreza and Rafiei (2023) analyzes vague questions and highlights issues in existing text-to-SQL benchmarks, where ambiguous questions are often linked to only one SQL query, leading to execution accuracy failures. AMBROSIA addresses this limitation by providing multiple SQL interpretations for ambiguous questions. Veltri et al. (2023) automatically generate declarative sentences containing facts that may lead to contradictions due to vague tables. Our approach, however, centers on human-written and verified questions that users might ask in real-world scenarios, rather than fact-checking.

Overall, our dataset introduces various types of ambiguity in questions, including scope and attachment ambiguities, which are often overlooked. Additionally, AMBROSIA features human-written interpretations and diverse SQL queries within a single ambiguity type and multi-table databases.

## 3 The AMBROSIA Dataset Creation

### 3.1 Formal Definition of Ambiguity

Before discussing how our dataset was created, we formally define ambiguity in text-to-SQL parsing, adapting the definition presented in Floratou et al. (2024):

**Definition 1.** Two SQL queries are **non-equivalent** if they produce different execution results, notwithstanding variations in layout or format.

**Definition 2.** Let $Q = \{q_1, \ldots, q_N\}$ denote the universe of non-equivalent SQL queries that can be formulated given a database $D$, with known database schema and values. Let $s$ denote a natural language question and $f : s \rightarrow P(Q)$ a function that operates in the context of database $D$ and deterministically maps $s$ to $P(Q)$, the power set of $Q$. Question $s$ is **ambiguous** if $f(s)$ has a cardinality of at least two.

This definition excludes ambiguities emanating from data management issues (e.g., relating to formatting, coverage, or the handling of NULL values), and assumes that the database schema and values are known. We also do not consider underspecification of the output format (e.g., whether the result should contain only specific columns or if auxiliary columns are acceptable). Instead, our focus is on ambiguity as a **linguistic phenomenon**, arising from the way a question is formulated, and leading to multiple interpretations and corresponding SQL queries. This ambiguity persists because the database context does not uniquely resolve the interpretations a question invites.

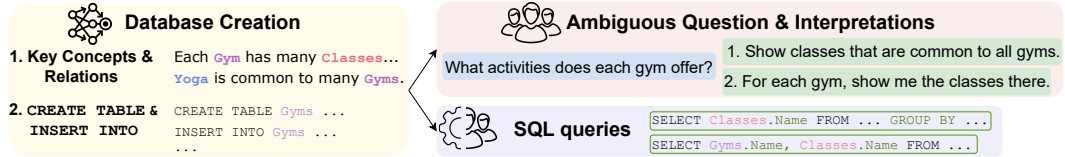

Figure 2: Annotation process for scope ambiguity in the "Health" domain.

## 3.2 Design Considerations

**Executable Logical Forms**    In designing our benchmark, we concentrated on text-to-SQL semantic parsing for several reasons. It represents a real-world use case where ambiguity arises naturally, e.g., in questions posed by users and the structure of databases. Unlike some other logical forms, SQL queries can be easily executed and inspected for correctness. The task is familiar to LLMs, they have demonstrated strong performance on standard benchmarks like Spider (Yu et al., 2018) and BIRD-Bench (Li et al., 2023b); it is reasonable to expect them to be able to parse SQL queries even in zero-shot settings since they likely have learned SQL syntax during training. This allows us to focus on ambiguity per se, rather than the model's ability to generate well-formed SQL.

**Databases that Support Ambiguity**    Another important consideration is ambiguity *in the context of a database*; it is not enough to just have ambiguous questions, they must also retain their ambiguity in relation to the database context. For instance, the question "What activities does each gym offer?" in Figure 1a is ambiguous precisely because there are fitness classes in the database common to multiple gyms (see the "Gyms_Classes" table). Most databases in academic text-to-SQL benchmarks (e.g., Spider) do not support ambiguous questions. As discussed earlier, modifying these databases, e.g., by adding tables or columns with synonymous names, makes them unrealistic with duplicate information and does not cover different types of ambiguity. In Section 3.3 we describe a controllable, multi-step approach that uses LLMs to generate databases supporting question ambiguity.

**Different Ambiguity Types**    Finally, we wish to include different types of ambiguity (see Figure 1).

*Scope ambiguity* arises when it is unclear which elements a quantifier, such as "each", "every", or "all", refers to. There are two possible interpretations for the ambiguous question in Figure 1a: in the collective interpretation, the quantifier is interpreted widely (i.e., "each gym" refers to all gyms in the database) and in the distributive interpretation the quantifier is interpreted narrowly (i.e., "each gym" is considered separately).

*Attachment ambiguity* occurs when it is unclear how a modifier or phrase is attached to the rest of the sentence. There are two possible interpretations for the question in Figure 1b: in the high attachment reading, the prepositional phrase "on a work-for-hire" is attached to the verb "show" (i.e., both screenwriters and editors are on work-for-hire contracts), whereas in the low attachment reading it is attached to "editors" (i.e., only editors have work-for-hire contracts, and screenwriters are on any contract). Within this category, we also consider attachment ambiguities for relative clauses (e.g., "writers and editors who have work-for-hire contracts") and adjectives (e.g., "work-for-hire editors and screenwriters") as their underlying database structure and SQL queries are similar to prepositional phrases.

*Vagueness* occurs when context creates uncertainty about which set of entities is being referred to. Similarly to ambiguous questions, there can be several interpretations. In the example in Figure 1c, the question has three interpretations depending on whether the answer refers to a general entity (e.g., the bank) or a more specific subtype (e.g., the branch), or both.

Scope and attachment ambiguities are well-known examples of structural ambiguity (Resnik, 1993; Kearns, 2000; Carnie, 2013; Kiss and Pafel, 2017) that arise when a sentence has more than one syntactic parse. However, the research community has only recently started exploring them in the context of LLMs (Liu et al., 2023a; Kamath et al., 2024a; Stengel-Eskin et al., 2024). We classify vagueness separately, as vague questions typically have a single syntactic parse, but, due to semantic imprecision, can refer to different database entities. We recognize that we do not exhaustively cover all cases of ambiguity in questions. For instance, we do not address lexical ambiguity (e.g., "Mississippi" as a river vs. state) which is less common in our context. We hope follow-on work will augment our dataset with additional types of ambiguity.

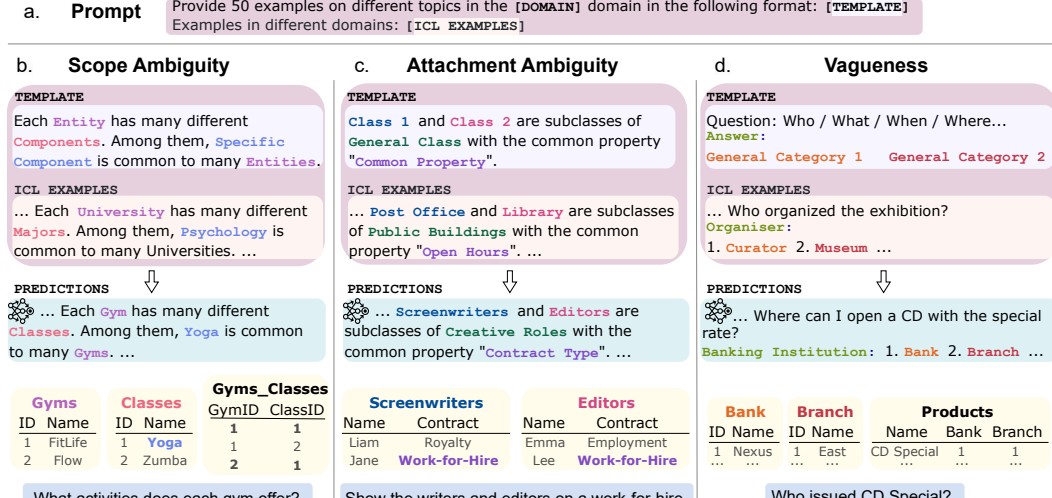

Figure 3: The prompt, templates, in-context examples (only one out of ten is shown for brevity, see Appendix C for the full versions), and predictions of key concepts and relations for each ambiguity type. Generated key concepts and relations later become sources of ambiguity in questions and databases (shown at the bottom for illustrative purposes).

We create ΑΜΒ ROSIA with these considerations in mind, following three steps: we select a domain and generate a database that supports one of the above ambiguity types; next, human annotators write ambiguous questions along with their unambiguous interpretations; finally, we automatically generate SQL queries based on templates for scope and attachment ambiguity, and ask annotators to write SQL queries for vague questions. Figure 2 shows a sketch of the annotation process for scope ambiguity.

## 3.3 Database Generation

In this section we describe the database generation process. We break this task into the following subtasks: selecting domains, generating key concepts and relations for each type of ambiguity, and generating CREATE TABLE and INSERT INTO SQL statements to build the database.

**Domains, Concepts, and Relations**    After analyzing existing semantic parsing datasets and publicly available databases, we compiled a list of real-world domains for database use. We refined this list to 16 domains (e.g., Entertainment, Banking, Hospitality; see the full list in Appendix B) as we found that some were either too narrow or too broad for LLM-based database generation.

To allow different interpretations in the context of a database, the latter must comply with constraints specific to a particular type of ambiguity. In the scope ambiguity example in Figure 1a, the database has information pertaining to different gyms and classes: a gym can offer multiple classes and a class can be offered by multiple gyms but one class, i.e., yoga, is common among them. Due to this structure (a many-to-many relationship with a common element), the question "What activities does each gym offer?" has two different interpretations (collective vs. distributive reading).

Similar to the conceptual data modeling stage used in database modeling, we first identify key concepts and relations that later become sources of ambiguity in questions and databases. For each type of ambiguity, we define a template that captures the general structure of the data and the potential for ambiguity. Using this template and ten in-context learning examples, we generate different structures (key concepts and relationships) within a given domain. Figure 3 shows the prompt, templates, in-context examples, and predictions which we obtain with a large language model (see Appendix C for the full prompts). Specifically, we employ OpenChat (Wang et al., 2024), one of the strongest open-sourced 7B LLMs with high performance in both code and text generation.

We manually inspect and filter LLM predictions with hallucinations or expressions otherwise unsuitable as database elements (e.g., proper nouns are unlikely to serve as table or column names). We found the vague category to be the most difficult to generate, requiring the most filtering. Examples

include entities unsuitable for further use as table or column names (e.g., "CGI" and "Practical Effects" can be valid answers for "What special effects were used in a movie?" but do not suit table or column names in a realistic database) or mutually exclusive (e.g., "Major Studio" and "Indie Producer" generally do not produce the same movie but are both types of production). Out of 1,883 predictions generated across all three categories, 881 (47%) were retained.

**Database Generation via SQL statements**    Based on the domain, key concepts, and relations from the previous step, we generate and execute `CREATE TABLE` statements to define database tables and their structure, as well as `INSERT INTO` statements to add values to tables. For this, we use the OpenChat model in a zero-shot setting. We determine possible database configurations for each type of ambiguity, since the concepts and relations generated in the previous step can be mapped to database elements in different ways. For instance, "Screenwriters" and "Editors" are two concepts predicted for the attachment example in Figure 3c, and can be used as table names (shown in the example) or column names within a single table. In total, we consider four configurations for databases that support attachment ambiguity, two for databases that support vagueness, and one for scope ambiguity, each of these corresponds to a separate instruction to the LLM (see details in Appendix D).

We first generate `CREATE TABLE` statements (4–6 tables per database on average), and then proceed to generate `INSERT INTO` statements (3–5 rows per table on average). At each step, we do not accept predictions that are non-executable or different from the selected configuration. We manually validate the generated databases and filter those we consider unnatural. For example, in Figure 3c, a database with "Contract Type" as a table would not be realistic, even though possible to generate. In contrast, "Projects", an alternative common property for screenwriters and editors, could be mapped to a column or table name. The total number of databases we obtain after filtering is 846.

## 3.4    Question and SQL Annotation

Ambiguous questions and their interpretations were written by human annotators. SQL queries for scope and attachment ambiguity were generated automatically, while those for vague questions were written by annotators. Annotators were recruited via the Prolific crowdsourcing platform based on various screening tasks; they were all native English speakers with prior knowledge of SQL or experience in database management. We manually reviewed all submissions and ensured all annotators followed our instructions. Details on crowdsourcing are provided in Appendix E.

**Scope and Attachment Ambiguity**    To create examples of scope and attachment ambiguity, we first automatically generate SQL queries using pre-defined templates for each database configuration (one for scope and four for attachment ambiguity). SQL queries are executed to ensure they yield non-empty, distinct results (and thus correspond to an ambiguous intent). We then automatically generate templates for questions (e.g., "What `Name` of `Classes` do we have for every `Gyms`?" in Figure 1a). Prolific annotators (20 in total) are shown a database as context and asked to write ambiguous questions and their interpretations in natural language based on these templates. The annotators substantially edited and paraphrased the original templates. The mean edit distance between our templates and their annotations is 9.2 for scope ambiguity and 12.3 for attachment ambiguity (higher values indicates greater deviation from the template, and edit distance is 0 when there is no difference).

In total, we obtained 501 questions with scope ambiguity, 362 questions with attachment ambiguity and two interpretations in natural language per question. Each ambiguous question corresponds to two golden SQL queries, and each interpretation is itself an unambiguous question and corresponds to one of these SQL queries.

**Vagueness**    A major difference between vague questions and those with scope and attachment ambiguities is that the former can map to very different SQL queries. To promote the diversity of SQL queries in our dataset, we ask experts (10 in total) to write SQL queries from scratch for this category rather than relying on predefined templates. To make the task easier, we simplify the databases by dropping and renaming tables or columns. For example, the "Bank" and "Branch" tables shown in Figure 3d are merged into a "Banking Institution" table (so that annotators inspect one general entity instead of two specific ones). Annotators write SQL queries and corresponding questions for the simpler tables (e.g., "Banking Institution") which we restore to the original databases once the annotation is complete. A question is ambiguous as an entity or its reference can be interpreted in different ways (e.g., a banking institution might be a bank, a branch, or both). From the SQL

Table 1: Comparison between AMBROSIA and other text-to-SQL datasets with ambiguous questions, NoisySP (Wang et al., 2023a) and AmbiQT (Bhaskar et al., 2023). # ambig and # unambig refer to the number of ambiguous questions and their unambiguous interpretations.

| | Scope | | Attachment | | Vague | | # DB | # Tab/DB |
| | # ambig | # unambig | # ambig | # unambig | # ambig | # unambig | | |
|---|---|---|---|---|---|---|---|---|
| NoisySP | 0 | 0 | 0 | 0 | 8,673 | 0 | 8,086 | 1.0 |
| AmbiQT | 0 | 0 | 0 | 0 | 23,295 | 0 | 200 | 5.1 |
| AMBROSIA | 501 | 1,002 | 362 | 724 | 414 | 1,239 | 846 | 5.0 |

query written by the annotators, we create SQL queries corresponding to different interpretations by replacing table or column names referring to the general entity with the specific original names. Again, we execute the queries to validate that they produce non-empty, distinct results.

Finally, a different batch of annotators (the same ones who wrote scope and attachment questions and interpretations) were shown vague questions and corresponding key concepts (e.g., a vague question "Who issued CD Special?" and key concepts "Bank" and "Branch" shown in Figure 3d) and asked to write interpretations in natural language. In total, we obtained 414 vague questions and 1,239 interpretations (two or three per question). Each question has two or three golden SQL queries, and each interpretation is itself an unambiguous question and corresponds to one of these SQL queries.

### 3.5 Dataset Analysis

Table 1 shows dataset statistics for AMBROSIA compared to two other text-to-SQL datasets with ambiguous questions (Wang et al., 2023a; Bhaskar et al., 2023). A unique aspect of AMBROSIA is that it includes three different types of ambiguity and provides interpretations in natural language for ambiguous questions. The number of tables per database in AMBROSIA is comparable to AmbiQT, which is based on Spider. Incidentally, our database generation approach could be used to augment existing text-to-SQL benchmarks, e.g., to assess robustness or out-of-domain generalization.

## 4 Experiments

Below we present an experimental framework for evaluating model performance on AMBROSIA and offer insights into model capabilities and failure modes. Implementation details are provided in Appendix F, and additional experimental results can be found in Appendix G.

**Models** We benchmark various large language models (LLMs) on AMBROSIA in light of their growing use and good performance on text-to-SQL tasks (Yu et al., 2018; Li et al., 2023b). Our experiments include LLMs of different sizes: **OpenChat-7B** (Wang et al., 2024), the model we used for database generation; the **instruction-tuned Llama3-8B** and **Llama3-70B** models from the Llama family (Dubey et al., 2024); **instruction-tuned CodeLlama-70B** (Roziere et al., 2023) which is trained specifically for code generation tasks; and **GPT-3.5 Turbo** and **GPT-4o** models from OpenAI.

**Prompting** AMBROSIA offers various options for exploring ambiguity in semantic parsing. In general, we expect a performant model to be able to recognize ambiguity in the context of a database and output as many SQL interpretations as applicable. Thus, our experiments follow two scenarios: (1) the model is given instructions that acknowledge the potential of ambiguity in the questions and specify that the output should include SQL queries for each possible interpretation; and (2) we provide standard text-to-SQL instructions but consider top-5 predictions from a beam of size 5 as does previous work (Bhaskar et al., 2023; Stengel-Eskin et al., 2024). We refer to the first method as **Prompt** and the second as **Beam**. In both scenarios, models have access to database context as we display the `CREATE TABLE` and `INSERT INTO` statements which fully describe the database schema and content. Databases in AMBROSIA do not have many rows, and as such fit within the context limits of the LLMs we use. We acknowledge that in real-world applications database content can be very large, requiring specific methods to extract related database entities. However, we leave this to future work. For Prompt, we conduct experiments with temperature equal to 0.5 and 5 random seeds and average the results for all models except for the OpenAI ones (due to cost constraints). For Beam, experimental results are deterministic since temperature is fixed to 0. We reserve 10% of the dataset for few-shot learning.

Table 2: Recall, precision and AllFound metrics for different zero-shot LLMs on AMBROSIA. Llama3-70B (Prompt) captures ambiguity best, with the highest recall on ambiguous questions. All models are unable to parse multiple interpretations when these exist (see AllFound metric).

| Model | Method | % Recall | | % Precision | | % AllFound |
| | | ambig | unambig | ambig | unambig | ambig |
| --- | --- | --- | --- | --- | --- | --- |
| OpenChat-7B | Prompt | 15.5 | 36.8 | 24.7 | 28.2 | 0.2 |
| | Beam | 14.7 | 37.9 | — | — | 1.1 |
| Llama3-8B | Prompt | 18.0 | 45.4 | 30.2 | 37.9 | 0.1 |
| | Beam | 19.9 | 48.6 | — | — | 1.7 |
| CodeLlama-70B | Prompt | 17.9 | 44.1 | 34.3 | 40.9 | 0.1 |
| | Beam | 25.4 | 56.2 | — | — | 0.1 |
| Llama3-70B | Prompt | **30.7** | 64.5 | 42.7 | 49.4 | **1.9** |
| | Beam | 28.0 | **65.5** | — | — | 1.4 |
| GPT-3.5 Turbo | Prompt | 26.7 | 61.6 | 40.2 | 52.1 | 0.5 |
| GPT-4o | Prompt | 27.1 | 63.4 | **51.1** | **59.6** | 0.4 |

**Evaluation Metrics**    A common approach to evaluating text-to-SQL semantic parsing is to compare whether the predicted SQL retrieves the same answer from the database as the gold logical form, typically by measuring execution accuracy. This method accommodates different formulations of the same SQL query and we employ it as well, but in our case, the output for ambiguous questions can be mapped to *several* correct SQL queries. We report recall and precision, but our primary focus is *recall* on *ambiguous* questions, as it captures the extent to which a model predicts different SQL queries that correspond to all possible interpretations. Following recent work (Bhaskar et al., 2023; Stengel-Eskin et al., 2024), we also measure whether *all* SQL queries are generated for ambiguous questions, i.e., whether recall is 100%. We call this metric AllFound.

## 4.1   Zero-Shot Results

Table 2 summarizes the performance of zero-shot models on AMBROSIA. We report results using micro-averaging. The standard deviation of the Prompt method is within 0.3%–1.2% for precision and recall for all models, except for CodeLlama, which varies from 4% to 5%. The standard deviation for AllFound is below 0.3%. For fairness, we do not report precision for the Beam method as it consistently outputs the top-5 predictions, although there are only 1–3 gold SQL queries.

As can be seen, all models demonstrate substantially higher recall on unambiguous questions compared to ambiguous ones, with differences ranging from 21% for OpenChat-7B to 36% for GPT-4o and 38% for Llama3-70B. As indicated by the AllFound metric, models generally fail to capture the ambiguity in the question as they rarely predict SQL queries for different interpretations. They often predict a correct SQL query for one interpretation only, which is why precision on ambiguous questions is higher than recall. Conversely, models sometimes predict more than one SQL query for unambiguous questions, which explains why they have lower precision than recall. Precision is 1.0 for both ambiguous and unambiguous questions when a model produces a *single* correct interpretation. However, when a model produces multiple predictions and only one is correct precision can drop significantly (e.g., to 0.1). This effect explains why ambiguous and unambiguous questions obtain somewhat similar precision.

Note that recall for unambiguous questions is the same as execution accuracy in standard text-to-SQL with one gold SQL query. Table 2 shows that the best recall is only 65.5% (achieved by Llama3-70B), which suggests AMBROSIA has challenging examples even in a standard text-to-SQL setting.

Overall, Llama3-70B (Prompt) captures ambiguity best, with the highest recall on ambiguous questions at nearly 31% and the highest AllFound value of 1.9% which is admittedly still very low. Llama3-70B performs better with the Prompt method, while CodeLlama-70B appears to perform better when considering top-k predictions (Beam), however, it never predicts both interpretations in this case. CodeLlama-70B is very unstable with the Prompt method, and shows performance comparable to Llama3-8B. This instability indicates that parsing ambiguous questions is significantly different from other code generation tasks encountered during training, requiring different skills (e.g., explicit instructions or documentation). GPT-4o performs best in terms of precision, however, even for this model, precision is lower than recall on unambiguous questions which means that it predicts SQL queries when it should not. **In general, all models fail to provide multiple SQL**

Table 3: Breakdown of model performance (zero-shot Llama3-70B) by ambiguity type shows that attachment ambiguity is most challenging.

| Prompt | % Recall ambig | unambig | % Precision ambig | unambig | % AllFound ambig |
|---|---|---|---|---|---|
| Scope | 41.5 | 90.4 | 52.7 | 66.4 | 2.9 |
| Attachment | 12.7 | 24.0 | 13.7 | 13.4 | 0.3 |
| Vague | 35.6 | 69.4 | 56.7 | 56.4 | 4.6 |

| Beam | % Recall ambig | unambig | % AllFound ambig |
|---|---|---|---|
| Scope | 41.6 | 91.8 | 1.1 |
| Attachment | 10.3 | 22.2 | 0.0 |
| Vague | 25.8 | 69.1 | 0.3 |

Table 4: Distribution of model predictions (zero-shot Llama3-70B) by interpretation type. Across ambiguities, there is a clear bias towards one interpretation type.

| Method | Scope Collective | Distributive | Attachment High | Low | Vague Component | Full |
|---|---|---|---|---|---|---|
| Prompt | 16.5 | 83.5 | 98.2 | 1.8 | 83.5 | 16.5 |
| Beam | 18.6 | 81.4 | 97.3 | 2.7 | 74.3 | 25.7 |

**queries when several interpretations are possible due to ambiguity but can mistakenly offer more than one SQL query for unambiguous questions.**

## 4.2 Analysis

We perform a more detailed analysis on the zero-shot instruction-tuned Llama 3-70B model, which has demonstrated the highest performance on our dataset. Table 3 presents recall, precision, and AllFound metrics for different categories of ambiguous and unambiguous questions. We observe that attachment ambiguity is most challenging. This category involves complex SQL queries (often requiring the UNION operator) and has the largest diversity in database configurations and corresponding gold SQL queries, which explains poor performance even on unambiguous questions. In contrast, we obtain best results for scope ambiguity, achieving nearly 92% recall (i.e., execution accuracy) on unambiguous questions, and 42% on ambiguous ones. We consider only one database configuration for scope ambiguity, which might be more familiar to LLMs due to the widespread use of many-to-many relationships. Interestingly, the model captures vagueness better with the Prompt method as evidenced by superior recall and AllFound results.

Table 4 shows the distribution of model predictions by interpretation type. We consider ambiguous examples that have at least one correctly predicted SQL query. For vague questions, we focus on those with three interpretations and define two types of predictions: those with only one component (interpretations 1 and 2 in Figure 1c; Component) and those with all components (interpretation 3 in Figure 1c; Full). There is a clear bias towards one interpretation type. The distributive interpretation is preferred for scope ambiguity (in 80% of cases) corroborating the findings of Kamath et al. (2024b). High attachment is chosen in more than 97% of cases for attachment ambiguity, and interpretations involving individual components are also overwhelmingly preferred in the case of vagueness. Overall, the Prompt method leads to more biased predictions compared to Beam.

## 4.3 Few-Shot Results

Figure 4 shows the performance of Llama3-70B (Prompt) in a few-shot learning setting. We select in-context examples randomly, each including ambiguous questions, their unambiguous interpretations, and corresponding SQL queries. We observe largest improvements in recall and precision over the zero-shot method with one to three examples. Recall on ambiguous questions improves by 4% but remains substantially worse compared to unambiguous questions. Increasing the number of examples helps, but improvements are not statistically significant given the 2–7% standard deviation.

Table 5 presents results with GPT-4o in one-shot setting (with 3 seeds). As can be seen, LLaMa3-70B and GPT-4o perform similarly: one-shot improves recall and AllFound in ambiguous questions for both models. The only difference is a slight (not statistically significant) decrease for LLaMa3-70B in precision, which could be due to the model generating more SQL queries, leading to more incorrect predictions and thus lower precision.

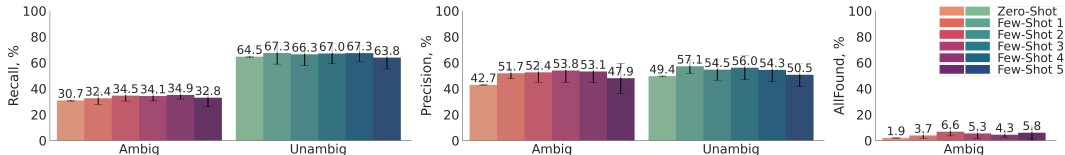

Figure 4: Recall, precision, and AllFound metrics for zero-shot and few-shot Llama3-70B. In-context examples are selected randomly. We obtain best results with 1-3 examples.

Table 5: Recall, precision, and AllFound metrics for Llama3-70B and GPT-4o in zero- and one-shot settings. We also include results for Llama3-70B simultaneously prompted with all three ambiguities and their definitions. Differences between models and settings are negligible.

| Model | ICL Examples | % Recall | | % Precision | | % AllFound |
| | | ambig | unambig | ambig | unambig | ambig |
|---|---|---|---|---|---|---|
| Llama3-70B | 0-shot | 30.7 | 64.5 | 42.7 | 49.4 | 1.9 |
| | 1-shot | 32.4 | **67.3** | **51.7** | 57.1 | 3.7 |
| | 3 ambiguities | **35.0** | 66.6 | 48.3 | 50.7 | 3.0 |
| GPT-4o | 0-shot | 27.1 | 63.4 | 51.1 | **59.6** | 0.4 |
| | 1-shot | 31.3 | 63.8 | 49.8 | 53.1 | **4.5** |

Further analysis indicates that providing more examples for one type of ambiguity improves performance for that type but may negatively impact others. We thus benchmark Llama3-70B with a prompt that includes *all* three ambiguities and their definitions from Section 3.2, unlike random sampling. The two prompt formats yield similar results, however, we observe a smaller standard deviation (2–4%) when prompting with all three ambiguities. Since it is unrealistic to have examples for all possible ambiguities, we consider these results an upper bound.

## 5 Limitations

Despite our best efforts to create a high-quality dataset, we cannot guarantee that AMBROSIA is error-free. Recall that we rely on annotators to provide ambiguous questions and their interpretations, both of which may have flaws. Hence, some interpretations may be unclear, failing to disambiguate the question, or unnatural and overly explicit with direct mentions to database entities. Our databases generally have simple and clear names, whereas in reality, they might be incomplete, have abbreviations, and so on. Since our experiments show that LLMs struggle to detect ambiguity and provide interpretations, we believe the current databases are well-suited for our task. However, future work might include augmentations to render them more realistic. When conducting experiments, we display the full database content, which is neither scalable nor safe for real-world applications. Consequently, our results can be seen as an upper bound on semantic parsing performance with ambiguous questions. Although our work broadens the scope of linguistic ambiguity in the text-to-SQL task, we acknowledge our dataset does not exhaustively cover all cases of ambiguity. We hope follow-on work will explore these further.

## 6 Conclusion

In this paper, we present AMBROSIA, a novel dataset for parsing ambiguous questions into SQL database queries across multiple domains. We populate AMBROSIA with multi-table realistic databases that support ambiguity, having developed an automatic pipeline for controlled database generation using key concepts and relations. AMBROSIA covers three types of ambiguity and contains ambiguous questions along with their interpretations in natural language. Our experiments demonstrate that even the most advanced LLMs struggle to capture ambiguity and provide accurate SQL queries for different interpretations of ambiguous questions, leaving ample room for improvement. We further hope AMBROSIA will spur future research on generalization (e.g., across domains and ambiguity types). Databases with fixed structures can be also modified to explore other interesting scenarios, including cases where the database context helps clarify originally ambiguous questions.

# 7 Acknowledgments

We thank the anonymous reviewers for their constructive feedback and Tom Hosking for his insightful comments. We gratefully acknowledge the support of the UK Engineering and Physical Sciences Research Council (grant EP/W002876/1).

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

## A Hosting, Licensing and Maintenance

The data, instructions, and code used for the creation of AMBROSIA and our benchmarking experiments are publicly available under the CC BY 4.0 license: `ambrosia-benchmark.github.io`. We plan to update the dataset by correcting any annotation errors as they are identified. A datasheet (Gebru et al., 2021) for AMBROSIA provides detailed documentation in Appendix H.

## B Domains

The dataset includes the following domains across all categories of ambiguity: Airport, Banking, Education, Entertainment, Filmmaking, Hospitality, Job Postings, and Journalism. Additionally, the Scope and Vague categories include the domains of Agriculture, Traffic, Streaming Services, Healthcare, and the Housing Market. The Vague category alone also encompasses the domains of Construction, Demographics, and Students.

## C Prompts for Key Concepts and Relations Generation

Below, we provide the prompts used for generating key concepts and relations. The domain is indicated in grey to be filled in with a specific name. We sample with a temperature of 0.6 and top_p of 0.95 until at least 35 concepts and relations are generated, or the number of attempts exceeds 5.

---

**Scope**: Key Concepts and Relations Generation

Provide 50 examples on different topics in the `DOMAIN` domain in the following format:

Each [Entity] has many different [Components]. Among them, [Specific Component] is common to many [Entities].

Examples in different domains:

1. Each `University` has many different `Majors`. Among them, `Psychology` is common to many `Universities`.
2. Each `Report` has many different `Sections`. Among them, `Introduction` is common to many `Reports`.
3. Each `Hospital` has many different `Amenities`. Among them, `Waiting Room` is common to many `Hospitals`.
4. Each `Musical` has many different `Roles`. Among them, `Narrator` is common to many `Musicals`.
5. Each `Phone` has many different `Features`. Among them, `Touchscreen` is common to many `Phones`.
6. Each `Museum` has many different `Events`. Among them, `Family Day` is common to many `Museums`.
7. Each `Company` has many different `Departments`. Among them, `Human Resources` is common to many `Companies`.
8. Each `Website` has many different `Pages`. Among them, `Homepage` is common to many `Websites`.
9. Each `Restaurant` has many different `Dishes`. Among them, `Pizza` is common to many `Restaurants`.
10. Each `Route` has many different `Stops`. Among them, `Transit Hub` is common to many `Routes`.

Examples in the `DOMAIN` domain:

---

**Attachment**: Key Concepts and Relations Generation

Provide 50 examples on different topics in the `DOMAIN` domain in the following format:

[Class 1] and [Class 2] are subclasses of [General Class]. All [Entities of Class 1] and [Entities of Class 2] have property "[Common Property]". There might be a [Entity of Class 1] and a [Entity of Class 2] that both have "[Common Property]" equal to "[Common Value]".

Examples in different domains:

---

1. `Post Office` and `Library` are subclasses of `Public Buildings`. All `Post Offices` and `Libraries` have the property "`Open Hours`". There might be a `Post Office` and a `Library` that both have "`Open Hours`" equal to "`8 a.m.`".

2. `Teacher` and `Lawyer` are subclasses of `Professional Occupations`. All `Teachers` and `Lawyers` have the property "`Education Level`". There might be a `Teacher` and a `Lawyer` that both have "`Education Level`" equal to "`Master's Degree`".

3. `Ballet` and `Musical` are subclasses of `Performing Arts`. All `Ballets` and `Musicals` have the property "`Performance Venue`". There might be a `Ballet` and a `Musical` that both have "`Performance Venue`" equal to "`Broadway Theater`".

4. `Apartment` and `Townhouse` are subclasses of `Residences`. All `Apartments` and `Townhouses` have the property "`Living Space Features`". There might be an `Apartment` and a `Townhouse` that both have "`Living Space Features`" equal to "`Balcony`".

5. `Bus` and `Train` are subclasses of `Public Transport`. All `Buses` and `Trains` have the property "`Scheduled Routes`". There might be a `Bus` and a `Train` that both have "`Scheduled Routes`" equal to "`Downtown Loop`".

6. `Hospital` and `Ambulatory Surgery Center` are subclasses of `Healthcare Facilities`. All `Hospitals` and `Ambulatory Surgery Centers` have the property "`Number of Patients`". There might be a `Hospital` and an `Ambulatory Surgery Center` that both have "`Number of Patients`" equal to "`Over 100 daily`".

7. `Thesis` and `Report` are subclasses of `Academic Writing`. All `Theses` and `Reports` have the property "`Research Focus`". There might be a `Thesis` and a `Report` that both have "`Research Focus`" equal to "`Renewable Energy`".

8. `Smartphone` and `Tablet` are subclasses of `Electronic Devices`. All `Smartphones` and `Tablets` have the property "`Charger Type`". There might be a `Smartphone` and a `Tablet` that both have "`Charger Type`" equal to "`USB-C`".

9. `Documentary` and `Animation` are subclasses of `Film Genres`. All `Documentaries` and `Animations` have the property "`Target Audience`". There might be a `Documentary` and an `Animation` that both have "`Target Audience`" equal to "`Young Adults`".

10. `Passport` and `Driving License` are subclasses of `Identification Documents`. All `Passports` and `Driving Licenses` have the property "`Date of Issue`". There might be a `Passport` and a `Driving License` that both have "`Date of Issue`" equal to "`June 1, 2018`".

Examples in the `DOMAIN` domain:

---

**Vague**: Key Concepts and Relations Generation

Provide 50 examples on different topics in the `DOMAIN` domain in the following format:

Question: [Who / What / How / When / Where ...]?  Subject of Inquiry: [Subject]
Focus: [Focus]  Possible answer types: 1. [General Category 1] 2. [General Category 2]

Examples in different domains:

1. Question: Who organized the exhibition?  Subject of Inquiry: `Exhibitions`
   Focus: `Organizer`  Possible answer types: 1. `Curator` 2. `Museum`

2. Question: When was this artefact created?  Subject of Inquiry: `Artefacts`
   Focus: `Timeframe`  Possible answer types: 1. `Historical Period` 2. `Estimated Age`

3. Question: What is the capacity of this venue?  Subject of Inquiry: `Venues`
   Focus: `Capacity`  Possible answer types: 1. `Seating Capacity` 2. `Total Capacity`

4. Question: How was this research funded?  Subject of Inquiry: `Research`
   Focus: `Funding`  Possible answer types: 1. `Grant` 2. `Funding Organization`

5. Question: Where was this play published?  Subject of Inquiry: `Books`
   Focus: `Place of Origin`  Possible answer types: 1. `Magazine` 2. `Country`

6. Question: What powers this vehicle?  Subject of Inquiry: `Vehicles`
   Focus: `Propulsion Method`  Possible answer types: 1. `Engine Type` 2. `Energy Source`

7. Question: When was this phone introduced?  Subject of Inquiry: `Electronics`

Focus: `Timeframe`     Possible answer types: 1. `Presentation Day`  2. `Release Date`

8. Question: Who won this contest?     Subject of Inquiry: `Contests`
   Focus: `Winner`     Possible answer types: 1. `Singer`  2. `Nation`

9. Question: What are the main features of this region?     Subject of Inquiry: `Regions`
   Focus: `Features`     Possible answer types: 1. `Landscape`  2. `Climate`

10. Question: Where is this painting displayed?     Subject of Inquiry: `Paintings`
    Focus: `Location`     Possible answer types: 1. `Gallery`  2. `City`

Examples in the `DOMAIN` domain:

# D   Database Generation

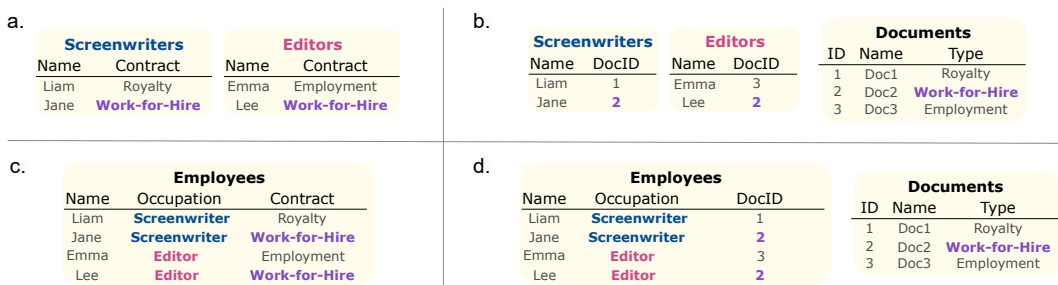

Figure 5: Database configurations that support attachment ambiguity.

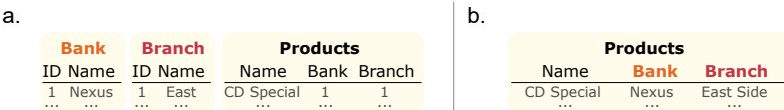

Figure 6: Database configurations that support vagueness.

Figure 5 shows four possible configurations for attachment ambiguity, each of them corresponding to concepts in Figure 3c and supporting the ambiguous request "Show the writers and editors on work for hire". Figure 6 shows two possible configurations for vague questions, each of them corresponding to concepts in Figure 3d and supporting the ambiguous request "Who issued CD Special?".

Below we provide examples of the prompts used for the generation of `CREATE TABLE` and `INSERT INTO` SQL statements for different ambiguity types and configurations. We indicate placeholders we substitute with specific values in grey. For databases with attachment and vague ambiguities, we provide the prompts corresponding to the configurations shown in Figures 5a and 6a, for the sake of brevity.

Each configuration has strict conditions we can validate when executing predicted SQL statements. For instance, in the case of scope ambiguity, we automatically discard a database if there is no connection between the "`Entity`" and "`Components`" tables. However, we relaxed some restrictions to generate more efficiently, as they can be satisfied by automatic modifications later. For example, we require a value in "`Components`" to be connected to multiple, but not necessarily all, values in "`Entity`," as we can later add the missing connections.

**Scope**: `CREATE TABLE`

Create multiple connected tables in the `DOMAIN` domain via SQLite, including a "`Entity`" table, a "`Components`" table (values: "`Specific_Component`" and others), and a "`Entities_Components`" table that joins these two tables (multiple rows of "`Entity`" are connected to "`Specific_Component`" and other values of "`Components`").

Other tables and columns are arbitrary. Each table must contain at least 3 columns.

Provide multiple `CREATE TABLE` statements in SQLite.

---

**Scope**: `INSERT INTO`

SQLite database in the `DOMAIN` domain:

`CREATE TABLE statements`

Insert 5 or more rows into each table.

"`Components`" table must contain "`Specific_Component`" and other values. All rows of table "`Entity`" must be linked to two rows of "`Components`", with "`Specific_Component`" and with another value, through "`Entities_Components`".

Provide 5 or more `INSERT INTO` statements for each table in SQLite.

---

**Attachment (a.)**: `CREATE TABLE`

Create multiple connected tables in the `DOMAIN` domain via SQLite. One table contains information about "`Class_1`" and "`Common_Property`" (containing values such as "`Common_Value`", etc.), and the other table contains information about "`Class_2`" and "`Common_Property`".

Other tables and columns are arbitrary. Each table must contain at least 3 columns.

Provide multiple `CREATE TABLE` statements in SQLite.

---

**Attachment (a.)**: `INSERT INTO`

SQLite database in the `DOMAIN` domain:

`CREATE TABLE statements`

Insert 5 or more rows into each table.

There must be:

- One row with "`Class_1`" and "`Common_Property`" equal to "`Common_Value`",
- One row with "`Class_1`" and "`Common_Property`" different from "`Common_Value`",
- One row with "`Class_2`" and "`Common_Property`" equal to "`Common_Value`",
- One row with "`Class_2`" and "`Common_Property`" different from "`Common_Value`",

Other rows are arbitrary. Use non-trivial values.

Provide 5 or more `INSERT INTO` statements for each table in SQLite.

---

**Vague (a.)**: `CREATE TABLE`

Create multiple connected tables in the `DOMAIN` domain via SQLite, including a table "`General_Category_1`", a table "`General_Category_2`" and a table "`Common_Subject`" connected to the tables "`General_Category_1`" and "`General_Category_2`".

Other tables and columns are arbitrary. Each table must contain at least 3 columns.

Provide multiple `CREATE TABLE` statements in SQLite.

---

**Vague (a.)**: `INSERT INTO`

SQLite database in the `DOMAIN` domain:

`CREATE TABLE statements`

Insert 5 or more rows into each table.

Provide 5 or more `INSERT INTO` statements for each table in SQLite.

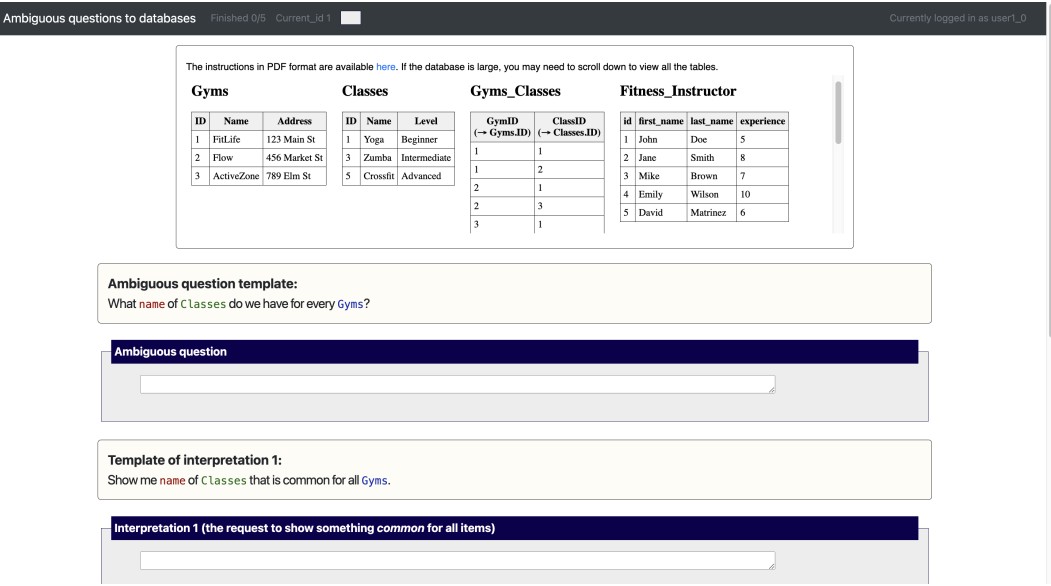

Figure 7: Annotation Interface: example of writing questions with scope ambiguity and their unambiguous interpretations.

# E Human Annotation

We recruited Prolific annotators using the following criteria: English as the first and primary language, no language-related disorders or literacy difficulties, knowledge of SQL and database management, a Prolific approval rate of 100%, and residency in the UK, USA, Ireland, Australia, New Zealand, or Canada. We also created trial tasks to choose the best annotators. Each annotator was paid £9 per hour. The total budget for annotation was £1,600. All submissions in Prolific are anonymous. All annotators participated voluntarily and had the option to return the task at any time if they chose not to complete it. We only collected data that was specifically requested from the annotators.

The first group of annotators (20 in total), participated in text-only tasks. These tasks included writing questions with scope and attachment ambiguity and unambiguous interpretations for all questions, including vague ones. We assigned 3–5 instances per person, which took 10–20 minutes to annotate. The second group (10 annotators in total), wrote SQL queries and corresponding questions that were later rendered vague. Each annotator completed 3 instances, taking an average of 20 minutes per instance.

We provided separate instructions with examples for each task and manually reviewed all submissions to ensure all annotators followed them. We elicited annotations using the Potato tool (Pei et al., 2022). An example of the interface is shown in Figure 7. The code and detailed instructions used for annotation are available at: `https://github.com/saparina/ambrosia`.

# F Implementation Details

**Models** We implemented the Prompt method using the Text Generation Inference toolkit[3] for the Llama-family models; for OpenChat (Wang et al., 2024), we used the official implementation provided by the authors and the OpenAI API for the GPT models[4]. We used the following versions of these models: openchat-3.5-1210, gpt-3.5-turbo-0125, gpt-4o-2024-05-13. The temperature for GPT models is 0. We implemented the Beam method using the vLLM library (Kwon et al., 2023). We used 4-8 CPUs, one NVIDIA-A100 GPU for OpenChat 7B and Llama3-8B and two NVIDIA-A100 GPUs for CodeLlama-70B and Llama3-70B. For the largest models, one zero-shot experiment (with fixed random seed) took one hour and one few-shot experiment with 5 examples took 5 hours.

---

[3]`https://github.com/huggingface/text-generation-inference`
[4]`https://platform.openai.com/`

**Evaluation** To compute evaluation metrics on model predictions, we represent the output of an SQL query as a set of values. This allows matching queries that correspond to the same interpretation and yield the same data even if they are structured differently, e.g., with different column orders or different operations. For instance, some gold queries in the attachment category use the UNION operator, but they could also be written using the JOIN operator. Although this relaxation might result in occasional false positives, we found it rarely happens as different SQL queries produce completely different outputs on our databases. We filter exact duplicates in predictions.

We measure precision as the number of correct queries in the output divided by the total number of predictions. Below we provide some examples:

Ambiguous case:

- If the model predicts only one SQL query from the gold set (one interpretation), precision is 1.0.
- If the model predicts all SQL queries from the gold set (all interpretations), precision remains 1.0.
- If multiple predictions include incorrect queries, precision falls between 0 and 1 (e.g., 0.5 might indicate one correct interpretation and one incorrect query, or two correct interpretations and two incorrect queries).

Unambiguous case:

- We have one gold SQL query. If the model predicts only this query, precision is 1.0.
- If the model predicts the correct query along with additional incorrect ones, precision is reduced (e.g., 1 / total number of predictions).

In both cases, if there are no correct queries, precision is 0. When there is only one correct prediction, precision is 1.

We measure recall as the number of correct queries in the output divided by the total number of ground truth queries (interpretations). Below we provide some examples:

Ambiguous case:

- If the model predicts all SQL queries from the ground truth set (covering all interpretations), recall is 1.0, even if additional incorrect queries are present in the output.
- If any ground truth interpretation is missing in the predictions, recall is between 0 and 1. For example, if there are two gold interpretations and the model predicts only one, recall would be 0.5.

Unambiguous case:

- If the model predicts the single gold query correctly, recall is 1.0, regardless of additional incorrect predictions.

In both cases, if there are no correct queries, recall is 0.

**Prompting** Below we provide the prompts used for the Prompt and Beam methods. We follow best practices for designing text-to-SQL prompts for LLMs, such as adding instructions like "Do not include any explanations, and do not select extra columns beyond those requested in the question" to prevent models from generating redundant columns or descriptions (Dong et al., 2023; Gao et al.). However, we found that the models do not always adhere to these instructions. To address this, we implemented a separate extractor for SQL queries from each model's output. Additionally, we observed that models often generate separate SQL queries for attachment ambiguity instead of a single complex query with a UNION of two subqueries. To resolve this, we added the instruction "Show the results in one table" to each question in this category.

Below we present the few-shot prompt with randomly selected in-context examples. Each example includes ambiguous questions, their unambiguous interpretations, and corresponding SQL queries:

**Few-Shot Prompt**

The task is to write SQL queries based on the provided questions in English. Questions can take the form of an instruction or command and can be ambiguous, meaning they can be interpreted in different ways. In such cases, write all possible SQL queries corresponding to different interpretations and separate each SQL query with an empty line. Do not include any explanations, and do not select extra columns beyond those requested in the question.

Some example databases, questions and corresponding SQL queries are provided based on similar problems:

Example 1:

Given the following SQLite database schema:

EXAMPLE_SQL_DATABASE_DUMP

Answer the following:
AMBIGUOUS_QUESTION

SQL query(s):
SQL_QUERY_1
SQL_QUERY_2
[OPTIONAL_SQL_QUERY_3]

Answer the following:
INTERPRETATION_1

SQL query(s):
SQL_QUERY_1

Answer the following:
INTERPRETATION_2

SQL query(s):
SQL_QUERY_2
. . .

Given the following SQLite database schema:

```
SQL_DATABASE_DUMP

Answer the following:
MAIN_QUESTION
```

Additionally, we consider the prompt with all three ambiguities and their definitions included:

---

**Few-Shot Prompt with 3 Ambiguities and Definitions**

The task is to write SQL queries based on the provided questions in English. Questions can take the form of an instruction or command and can be ambiguous, meaning they can be interpreted in different ways. In such cases, write all possible SQL queries corresponding to different interpretations and separate each SQL query with an empty line. Do not include any explanations, and do not select extra columns beyond those requested in the question.

Some example databases, questions and corresponding SQL queries are provided based on similar problems:

Type 1: Scope Ambiguity

Definition: Scope ambiguity issue arises when it is unclear which elements a quantifier, such as "each", "every" or "all", refers to.

Example:

Given the following SQLite database schema:

`SCOPE_SQL_DATABASE_DUMP`

Answer the following:
`SCOPE_QUESTION`

SQL query(s):
`SQL_QUERY_1`
`SQL_QUERY_2`

Answer the following:
`SCOPE_INTERPRETATION_1`

SQL query(s):
`SQL_QUERY_1`
. . .

Type 2: Attachment Ambiguity

Definition: Attachment ambiguity occurs when it is unclear how a modifier or phrase is attached to the rest of the sentence.

`ATTACHMENT_EXAMPLE`

Type 3: Vagueness

Definition: Vagueness occurs when context creates uncertainty about which set of entities is being referred to.

`VAGUE_EXAMPLE`

Given the following SQLite database schema:

`SQL_DATABASE_DUMP`

Answer the following:
`MAIN_QUESTION`

---

Table 6: Number of unique SQL queries predicted by zero-shot Llama3-70B (Prompt) and number of unique results after execution of these queries. The model predicts more than one query even for unambiguous questions.

| # Unique SQL Queries | | # Unique Exec Results | |
|---|---|---|---|
| ambig | unambig | ambig | unambig |
| 2.4 | 2.3 | 2.0 | 1.7 |

Table 7: Zero-shot accuracy of the Llama3-70B in detecting ambiguity in questions. The model overestimates ambiguity.

| % Accuracy | |
|---|---|
| ambig | unambig |
| 81.2 | 26.1 |

## G  Additional Results

**Extended Analysis**   Table 6 presents the number of unique SQL queries predicted by the zero-shot Llama3-70B (Prompt) and the number of unique results obtained upon executing these predictions. As can be seen, the number of execution results is consistently lower than the number of predicted SQL queries, indicating that some queries, despite differing in structure, yield the same results. Typically, both metrics are around 2, implying that the model is capable of generating multiple distinct SQL queries that produce different results. Although the figures for unambiguous questions are lower than for ambiguous ones, they still do not approach the optimum value of 1.0 (unambiguous questions correspond to a single SQL query).

To further explore how the model captures ambiguity, we consider the task of ambiguity detection. We present the model with a question and ask it to determine whether it is ambiguous. Similar to the Prompt method, we provide the model with comprehensive instructions regarding this task:

> **Ambiguity Detection**
>
> The task is to identify ambiguous questions in English that are intended to interact with an SQLite database. Questions can take the form of an instruction or command and can be ambiguous, meaning they can be interpreted in different ways (corresponding to different SQL queries that produce different results). Answer Yes or No, and do not include any explanations.
>
> Given the following SQLite database schema:
>
> SQL_DATABASE_DUMP
>
> Is the following question ambiguous:
> QUESTION

Table 7 shows the accuracy of zero-shot Llama3-70B in detecting ambiguity. Our results indicate that the model tends to overestimate ambiguity, classifying questions as ambiguous in approximately 80% of cases, regardless of whether they genuinely are. This is a limitation of prompt-based approaches, which may confuse the model rather than provide helpful guidance.

**Error Cases**   We conducted an error analysis on the zero-shot Llama3-70B (Prompt) by randomly selecting 50 examples where the model failed. Our analysis revealed errors typical of standard text-to-SQL parsing, as well as errors specifically related to ambiguity. Common text-to-SQL errors included: incorrect output columns (29%), including superfluous columns contrary to our instructions, errors in the filters and joins (17%), failure to include a UNION of two queries (17%, typical for the attachment category), and execution errors (5%). The most frequent error relating to ambiguity is the generation of different variations of the same SQL query, which occurred in 34% of cases, even for unambiguous questions. For ambiguous queries, the model provided only one interpretation in 11% of cases. See the examples provided next.

|  | **Type:** Scope | **Domain:** Agriculture | **Ambiguous?** Yes |
|---|---|---|---|

**Question**   What kind of crop does each farm offer?

**Predictions**   ✓ SELECT f.Name, c.CropName FROM Farms f JOIN FarmsCrops fc ON f.FarmID = fc.FarmID JOIN Crops c ON fc.CropID = c.CropID

✓ SELECT DISTINCT f.Name, c.CropName FROM Farms f JOIN FarmsCrops fc ON f.FarmID = fc.FarmID JOIN Crops c ON fc.CropID = c.CropID

**Gold**   1. SELECT c.CropName FROM FarmsCrops fc JOIN Crops c ON fc.CropID = c.CropID JOIN Farms f ON fc.FarmID = f.FranID GROUP BY fc.CropID HAVING COUNT(DISTINCT fc.FarmID) = (SELECT COUNT(*) FROM Farms)

2. SELECT f.name, c.cropName FROM FarmsCrops fc JOIN Crops c ON fc.CropID = c.CropID JOIN Farms f ON fc.FarmID = f.FarmID

**Comments**   All predicted queries are correct but they all provide the same information and correspond to one interpretation (distributive; gold query 2).

|  | **Type:** Scope | **Domain:** Healthcare | **Ambiguous?** No |
|---|---|---|---|

**Question**   For every program, list the activity descriptions

**Predictions**   ✗ SELECT hpp.description, a.description FROM Health_Promotion_Program hpp JOIN Health_Promotion_Program_Activities hppa ON hpp.program_id = hppa.program_id JOIN Activities a ON hppa.activity_id = a.activity_id

✗ SELECT a.description FROM Health_Promotion_Program hpp JOIN Health_Promotion_Program_Activities hppa ON hpp.program_id = hppa.program_id JOIN Activities a ON hppa.activity_id = a.activity_id GROUP BY hpp.program_id

**Gold**   SELECT hpp.name, a.description FROM Health_Promotion_Program_Activities hppa JOIN Activities a ON hppa.activity_id = a.activity_id JOIN Health_Promotion_Program hpp ON hppa.program_id = hpp.program_id

**Comments**   The first predicted query has an error in the output columns: it outputs the program description instead of the program name. The second query omits the program name column altogether.

|  | **Type:** Attachment | **Domain:** Filmmaking | **Ambiguous?** Yes |
|---|---|---|---|

**Question**   Provide all sound designs and music scores created by Thomas Newman. Show them in one table.

**Predictions**   ✗ SELECT description FROM SoundDesigns WHERE creatorId = (SELECT id FROM Creators WHERE name = 'Thomas Newman')

✗ SELECT description FROM MusicScores WHERE creatorId = (SELECT id FROM Creators WHERE name = 'Thomas Newman')

✓ SELECT sd.description, ms.description FROM SoundDesigns sd, MusicScores ms WHERE sd.creatorId = ms.creatorId AND sd.creatorId = (SELECT id FROM Creators WHERE name = 'Thomas Newman')

✓ SELECT * FROM (SELECT description FROM SoundDesigns WHERE creatorId = (SELECT id FROM Creators WHERE name = 'Thomas Newman')) UNION SELECT * FROM (SELECT description FROM MusicScores WHERE creatorId = (SELECT id FROM Creators WHERE name = 'Thomas Newman'))

**Gold**   1. SELECT sd.description FROM Creators c JOIN SoundDesigns sd ON c.id = sd.creatorId WHERE c.name = "Thomas Newman" UNION SELECT ms.description FROM Creators c JOIN MusicScores ms ON c.id = ms.creatorInterId WHERE c.name = "Thomas Newman"

2. SELECT sd.description FROM Creators c JOIN SoundDesigns sd ON c.id = sd.creatorId UNION SELECT ms.description FROM Creators c JOIN MusicScores ms ON c.id = ms.creatorId WHERE c.name = "Thomas Newman"

**Comments**   The first two predictions should be subqueries of a single complex query (failure to include a UNION of two queries). The last two queries are correct; however, they provide the same information and correspond to one interpretation (with high attachment; gold query 1).

|  |  |  |
|---|---|---|
| **Type:** Attachment | **Domain:** Hospitality | **Ambiguous?** No |

**Question**   Show all youth hostels and also show the hotels, but only those with 50-priced rooms. Show them in one table.

**Predictions**   ✗ `SELECT name, location, accommodation_type FROM hotels WHERE accommodation_type = 'Youth Hostel'`

✗ `SELECT name, location, accommodation_type FROM hotels WHERE id IN (SELECT hotel_id FROM rooms WHERE price = 50.0)`

✗ `SELECT name, location, accommodation_type FROM hotels WHERE accommodation_type = 'Youth Hostel' UNION SELECT name, location, accommodation_type FROM hotels WHERE id IN ( SELECT hotel_id FROM rooms WHERE price = 50.0 )`

**Gold**   `SELECT h.name FROM hotels h JOIN rooms r ON h.id = r.hotel_id WHERE h.accommodation_type = "Youth Hostel" OR (h.accommodation_type = "Hotel" AND r.price = 50.0)`

**Comments**   All predicted queries have redundant output columns. The first two predictions should be subqueries of a single complex query (failure to include a UNION of two queries).

|  |  |  |
|---|---|---|
| **Type:** Vague | **Domain:** Journalism | **Ambiguous?** Yes |

**Question**   List all the articles with their respective editor and author names.

**Predictions**   ✓ `SELECT a.title, me.name AS editor, au.fullname AS author FROM Articles a JOIN Final_Draft fd ON a.final_draft_id = fd.id JOIN Managing_Editor me ON fd.managing_editor_id = me.id JOIN Authors au ON a.author_id = au.author_id`

✓ `SELECT a.title, ce.name AS editor, au.fullname AS author FROM Articles a JOIN Final_Draft fd ON a.final_draft_id = fd.id JOIN Copy_Editor ce ON fd.copy_editor_id = ce.id JOIN Authors au ON a.author_id = au.author_id`

**Gold**   1. `SELECT A.title, ME.name, AU.fullname FROM Articles A JOIN Final_Draft FD ON FD.id = A.final_draft_id JOIN Managing_Editor ME ON ME.id = FD.managing_editor_id JOIN Authors AU ON AU.author_id = A.author_id`

2. `SELECT A.title, CE.name, AU.fullname FROM Articles A JOIN Final_Draft FD ON FD.id = A.final_draft_id JOIN Copy_Editor CE ON CE.id = FD.Copy_Editor_id JOIN Authors AU ON AU.author_id = A.author_id`

3. `SELECT A.title, ME.name, CE.name, AU.fullname FROM Articles A JOIN Final_Draft FD ON FD.id = A.final_draft_id JOIN Managing_Editor ME ON ME.id = FD.managing_editor_id JOIN Authors AU ON AU.author_id = A.author_id JOIN Copy_Editor CE ON CE.id = FD.Copy_Editor_id`

**Comments**   All predicted queries are correct and correspond to two different interpretations (Component), but the third interpretation (Full) is missing.

|  |  |  |
|---|---|---|
| **Type:** Vague | **Domain:** Student | **Ambiguous?** No |

**Question**   List students born after 1990, along with their field of study.

**Predictions**   ✗ `SELECT fullname, Field_Of_Study FROM Students WHERE BirthDate > '1990-01-01'`

✗ `SELECT fullname, Field_Of_Study FROM Students WHERE YEAR(BirthDate) > 1990`

**Gold**   `SELECT fullname, Field_Of_Study FROM Students WHERE CAST(SUBSTR(BirthDate, 1, 4) AS INT) > '1990'`

**Comments**   The first predicted query has an error in its filter as it incorrectly includes students born in 1990. The second query is not executable as SQLite does not have a built-in `YEAR` function.

# H Datasheet for AMBROSIA

Below we provide a datasheet (Gebru et al., 2021) for AMBROSIA.

## Motivation

**For what purpose was the dataset created?** AMBROSIA aims to study ambiguity in semantic parsing, with a focus on text-to-SQL parsing due to its real-world relevance. Large Language Models (LLMs) have demonstrated strong performance on existing unambiguous text-to-SQL benchmarks, allowing us to concentrate specifically on studying ambiguity. Unlike other logical forms, such as the first-order logic used by Stengel-Eskin et al. (2024), SQL queries are straightforward to execute and verify, making them ideal for testing. We aim to replicate practical semantic parsing scenarios with diverse databases in various domains that support ambiguity in questions. We also cover three different types of ambiguity and provide human-written interpretations that enable in-depth analysis. This distinguishes AMBROSIA from other benchmarks (Wang et al., 2023a; Bhaskar et al., 2023), which rely on augmentations of existing datasets, and are thus less realistic, and lack diversity.

**Who created the dataset (e.g., which team, research group) and on behalf of which entity (e.g., company, institution, organization)?** It was created by the authors.

**Who funded the creation of the dataset?** We gratefully acknowledge the support of the UK Engineering and Physical Sciences Research Council (grant EP/W002876/1).

## Composition

**What do the instances that comprise the dataset represent (e.g., documents, photos, people, countries)?** AMBROSIA includes databases, ambiguous questions and requests related to them, their corresponding interpretations and SQL queries.

**How many instances are there in total (of each type, if appropriate)?** AMBROSIA contains 846 multi-table databases in 16 distinct domains, 1,277 ambiguous questions, their unambiguous interpretations provided by humans and complex SQL queries (2,965 in total).

**Does the dataset contain all possible instances or is it a sample (not necessarily random) of instances from a larger set?** The dataset includes all instances we collected. Our data collection approach allows for future expansion of the dataset.

**What data does each instance consist of?** One example in AMBROSIA includes a database that supports ambiguity, an ambiguous question, its possible interpretations, and SQL queries corresponding to these interpretations. Interpretations can be viewed separately as unambiguous questions.

**Is there a label or target associated with each instance?** For each question, the targets are the corresponding SQL queries: one query for an unambiguous question and 2–3 queries for an ambiguous one. Additionally, each database is labeled based on the domain it represents, its configuration, and the type of ambiguity present in the relevant questions.

**Is any information missing from individual instances?** No

**Are relationships between individual instances made explicit (e.g., users' movie ratings, social network links)?** Annotators were given databases for context when writing questions, interpretations, and SQL queries for vague questions. Other SQL queries were constructed to query the given database. Annotators provided interpretations when they were aware of the ambiguous question they related to.

**Are there recommended data splits (e.g., training, development/validation, testing)?** We reserve 10% for a few-shot learning scenario and recommend to evaluate on the remaining data.

**Are there any errors, sources of noise, or redundancies in the dataset?** We manually validate all databases and human annotations. However, there might be annotation errors that we did not notice, such as interpretations that fail to disambiguate the questions or questions that are not ambiguous. The generated database might also contain tables that are irrelevant to the chosen key concepts.

**Is the dataset self-contained, or does it link to or otherwise rely on external resources (e.g., websites, tweets, other datasets)?** The dataset is self-contained.

**Does the dataset contain data that might be considered confidential (e.g., data that is protected by legal privilege or by doctor-patient confidentiality, data that includes the content of individuals' non-public communications)?** No.

**Does the dataset contain data that, if viewed directly, might be offensive, insulting, threatening, or might otherwise cause anxiety?** No.

**Does the dataset relate to people?** Yes.

**Does the dataset identify any subpopulations (e.g., by age, gender)?** There might be questions or databases containing information about age or gender, but they do not provide any opinions.

**Is it possible to identify individuals (i.e., one or more natural persons), either directly or indirectly (i.e., in combination with other data) from the dataset?** No, all annotation submissions were fully anonymous.

**Does the dataset contain data that might be considered sensitive in any way (e.g., data that reveals racial or ethnic origins, sexual orientations, religious beliefs, political opinions or `UNION` memberships, or locations; financial or health data; biometric or genetic data; forms of government identification, such as social security numbers; criminal history)?** We manually verify that the dataset does not contain any sensitive or harmful information.

**Collection Process**

**How was the data associated with each instance acquired?** Human annotators wrote ambiguous questions along with their unambiguous interpretations. Databases were generated by the LLM OpenChat ((Wang et al., 2024)), and SQL queries were automatically created using templates to address scope and attachment ambiguity. Annotators were asked to write SQL queries and corresponding questions for databases, which we had modified by merging vague concepts into more general ones, thereby removing ambiguity. After the annotation was complete, we restored the original databases and adjusted the SQL queries. The questions became vague due to the reintroduction of vague concepts. We manually verified questions, their interpretations and databases. SQL queries were executed to validate that they produced non-empty, distinct results. See details in Section 3 and Appendix E.

**What mechanisms or procedures were used to collect the data (e.g., hardware apparatus or sensor, manual human curation, software program, software API)?** We elicited annotations using the Potato tool (Pei et al., 2022).

**If the dataset is a sample from a larger set, what was the sampling strategy (e.g., deterministic, probabilistic with specific sampling probabilities)?** N/A

**Who was involved in the data collection process (e.g., students, crowdworkers, contractors) and how were they compensated (e.g., how much were crowdworkers paid)?** We recruited annotators through the Prolific crowdsourcing platform. Each annotator was paid £9 per hour.

**Over what timeframe was the data collected?** October 2023–April 2024.

**Were any ethical review processes conducted (e.g., by an institutional review board)?** No.

**Does the dataset relate to people?**    Yes.

**Did you collect the data from the individuals in question directly, or obtain it via third parties or other sources (e.g., websites)?**    The data was collected directly.

**Were the individuals in question notified about the data collection?**    Yes, by requirement process in Prolific.

**Did the individuals in question consent to the collection and use of their data?**    Yes, by requirement process in Prolific.

**If consent was obtained, were the consenting individuals provided with a mechanism to revoke their consent in the future or for certain uses?**    No.

**Has an analysis of the potential impact of the dataset and its use on data subjects (e.g., a data protection impact analysis) been conducted?**    No, all submissions on Prolific are anonymous, and we did not collect any data beyond what was requested.

**Preprocessing/Cleaning/Labeling**

**Was any preprocessing/cleaning/labeling of the data done (e.g., discretization or bucketing, tokenization, part-of-speech tagging, SIFT feature extraction, removal of instances, processing of missing values)?**    We manually validated all generated concepts and databases and human-written submissions.

**Was the "raw" data saved in addition to the preprocessed/cleaned/labeled data (e.g., to support unanticipated future uses)?**    No.

**Is the software used to preprocess/clean/label the instances available?**    Yes, see `https://github.com/saparina/ambrosia`

**Uses**

**Has the dataset been used for any tasks already?**    Yes, we benchmarked LLMs on the text-to-SQL semantic parsing with ambiguous questions.

**Is there a repository that links to any or all papers or systems that use the dataset?**    No.

**What (other) tasks could the dataset be used for?**    It can be used for studying different types of generalization, e.g., across domains or types of questions.

**Is there anything about the composition of the dataset or the way it was collected and preprocessed/cleaned/labeled that might impact future uses?**    We cannot guarantee that the dataset is free from errors (see Section 5 for limitations). Additionally, the recruitment of annotators with knowledge of database management and SQL introduces a potential bias. Future research could mitigate this by paraphrasing the questions.

**Are there tasks for which the dataset should not be used?**    No.

**Distribution**

**Will the dataset be distributed to third parties outside of the entity (e.g., company, institution, organization) on behalf of which the dataset was created?**    Yes, the dataset is publicly available.

**How will the dataset be distributed (e.g., tarball on website, API, GitHub)?**    The dataset is available at `ambrosia-benchmark.github.io`

**When will the dataset be distributed?**    Now.

**Will the dataset be distributed under a copyright or other intellectual property (IP) license, and/or under applicable terms of use (ToU)?** The dataset is distributed under CC BY 4.0

**Have any third parties imposed IP-based or other restrictions on the data associated with the instances?** No.

**Do any export controls or other regulatory restrictions apply to the dataset or to individual instances?** No.

**Maintenance**

**Who is supporting/hosting/maintaining the dataset?** The authors of this paper.

**How can the owner/curator/manager of the dataset be contacted (e.g., email address)?** The first author of this paper.

**Is there an erratum?** No.

**Will the dataset be updated (e.g., to correct labeling errors, add new instances, delete instances)?** We plan to update the dataset by correcting any annotation errors as they are identified. The new version will be labeled accordingly.

**If the dataset relates to people, are there applicable limits on the retention of the data associated with the instances (e.g., were individuals in question told that their data would be retained for a fixed period of time and then deleted)?** No.

**Will older versions of the dataset continue to be supported/hosted/maintained?** No.

**If others want to extend/augment/build on/contribute to the dataset, is there a mechanism for them to do so?** Yes, the code used for data collection is available at `https://github.com/saparina/ambrosia`.

