# OpenReview forum: "AMBROSIA: A Benchmark for Parsing Ambiguous Questions into Database Queries"
_NeurIPS.cc/2024/Datasets_and_Benchmarks_Track — NeurIPS 2024 Track Datasets and Benchmarks Spotlight_

### Official Review · Reviewer_zcDZ · 2024-07-22
**All my concerns are about the definitions of 3 ambiguity types and whether the benchmark can be tackled in a trivial few-shot prompting way.**

**Rating:** 7
**Confidence:** 4
**Correctness:** Yes

**Review:**

The ambiguity in text-to-SQL is very attractive to me. On one hand, it is a very common phenomenon in the real world, since users often do not convey their intention clearly. On another hand, existing work mostly focus on morphological variations on existing DBs which might   be unrealistic and trivial.

Pros: see Strengths.

Cons: see Opportunities For Improvement

**Strengths:**

1. This work inspects text-to-SQL ambiguity in multi-table settings. It focuses on more realistic settings rather than simple modification on DB schema items.

2. It proposes a thorough LLM-driven framework to collect ambiguous examples which might inspire the dataset construction procedure in other specialized or interesting downstream tasks.

**Additional Feedback:**

None

**Clarity:**

Yes.

About the metric precision, I am a little confused. How do you measure the precision since for ambiguous questions, there are multiple solutions. If the execution results of any output SQL are correct with the ground-truth, we consider precision=1.0 ? It seems the precision for ambiguous and unambiguous questions are almost at the same level. Therefore, a more detailed explanation is vital.

**Documentation:**

Yes

**Limitations:**

Yes. The limitations are mostly about potential errors in annotation and mismatches with realistic settings (table/column name abbrev., and large DB contents). I am open with these issues and they are not the main focuses of this work.

**Opportunities For Improvement:**

1. Are these three ambiguities complete and mutually exclusive? Did you just come up with this on your own or do you have any basis for it? This is my major concern of this work: there lacks a formal (not the paragraph in line 103, I mean some other linguistic references or authority sources) and comprehensive (perhaps 3 ambiguity types are still narrow and not complete) definition of ambiguity.

2. In line 99, the author mentions that "modifying these databases, e.g., by adding tables or columns with synonymous names, makes them unrealistic". However, use LLMs to generate could also introduce some bias in output distribution and cause unreality.

3. Another major concern is that, test data (or the benchmark) involves a significant participation of LLMs with pre-defined templates. There exists an evident underlying pattern caused by this standardized procedure, which might be easily captured by inference models. After all, there are merely 3 types of ambiguities (see item 1) and LLM predictions have potential output biases (see item 2). I heavily suspect that, **if we inject prompts about concrete definitions of these three ambiguities and one case plus interpretations+solutions for each will dramatically boost the eventual performances**. For example, in the task prompt, we can insert the following text:

######################

We enumerate 3 types of ambiguities with examples for you.

Type1: scope ambiguity

Definition: this issue arises when it is unclear which elements a quantifier ...

Here is one example:

Question: xxx

Database schema: xxx

Interpretation 1: xxx

SQL 1: xxx

Interpretation 2: xxx

SQL 2: xxx


Type 2: attachment ambiguity

...

######################

I notice that in Section 4.3, you provide the few-shot results without prompting details. What is the syste, task, and exemplar prompt? I only found zero-shot prompt in line 565. Did you include definitions and cases of all 3 types of ambiguity questions with interpretations? Could you provide the results of more advanced LLMs in few-shot settings since closed-source models still outperforms open-sourced LLaMa3-70B by a large margin?

These are just my assumption, and I am willing to be convinced if you can demonstrate that this simple few-shot prompting strategy will not work, such that the proposed benchmark can not be tackled in a trivial way.

4. Typos: in Table 1, # of questions/interpretations for Attachment and Vague are not consistent with statements in line 187 and 206.

**Relation To Prior Work:**

Yes

**Summary And Contributions:**

This work focuses on text-to-SQL tasks with intrinsic ambiguity. It proposes a text-to-SQL benchmark "AMBROSIA", which covers 16 domains, involves 849 DBs, 4240 ambiguous SQL quries. It mentions three types of ambiguity, scope ambiguity, attachment ambiguity, and vagueness. And authors propose a complete dataset curation procedure and perform experiments on the dataset to demonstrate existing LLMs (even GPT-4o) perform terribly in identifying and tackling these 3 types of ambiguities.

---

> ### Author Rebuttal · Authors · 2024-08-16
>
> **1. Ambiguity definitions**
>
> > Are these three ambiguities complete and mutually exclusive? ... This is my major concern of this work: there lacks a formal and comprehensive definition of ambiguity.
>
> Thank you for the useful feedback! We provide a formal definition of ambiguity in General Response.
>
> Our study focuses on ambiguity as a **linguistic phenomenon** within the text-to-SQL task. Scope and attachment ambiguities are well-known examples of structural ambiguity (Resnik, 1993; Kearns, 2011; Carnie, 2013; Kiss and Pafel, 2017) that arise when a sentence has more than one syntactic parse. These types of ambiguities have been explored in the context of semantic parsing (Stengel-Eskin et al., 2024). Within our dataset, attachment ambiguity includes subtypes such as prepositional phrase attachment, relative clause ambiguities, and adjective ambiguities (L114-116).
>
> We classify vagueness separately, as vague questions typically have a single syntactic parse but, due to semantic imprecision, can refer to different database entities. For example, the scenario shown in Figure 1c involves specific vs. non-specific reference ambiguity (Kearns, 2011), but there are also other cases like “Presentation Day/Release Day.”
>
> We recognize that, even when considering ambiguity only in questions, there are more types that we do not cover. Additionally, we intentionally excluded combinations of different ambiguities, ensuring they do not appear in the dataset, as this is a more advanced problem that seems too challenging given the current model performance on our benchmark. However, our dataset collection procedure could be extended to include such mixed cases, which we plan to explore in future work.
>
> We will include these clarifications and the relevant literature in the final version of the paper.
>
> ---
> *Resnik, 1993. Semantic Classes and Syntactic Ambiguity. HLT 1993.*
>
> *Kearns, 2011. Semantics.*
>
> *Carnie, 2013. Syntax: A Generative Introduction.*
>
> *Kiss and Pafel, 2017. Quantifier Scope Ambiguities.*
>
> **2. Few-Shot prompting**
> > Test data involves a significant participation of LLMs with pre-defined templates.. which might be easily captured by inference models… I heavily suspect that, if we inject prompts about concrete definitions of these three ambiguities and one case plus interpretations+solutions for each will dramatically boost the eventual performances.
>
> Showing all three ambiguity types together is an alternative few-shot setup. We follow the prompt format suggested by the reviewer and report the following results for Llama3-70B:
> | Prompt |% Recall (Ambig)|% Recall (Unambig)|% Precision (Ambig)|% Precision (Unambig)|% AllFound|
> |---|---|---|---|---|---|
> |3 different examples + definitions|39.6|73.7|53.8|54.3|5.0|
> |1-shot|38.9|73.8|55.6|58.7|5.4|
> |zero-shot|29.9|57.5|41.2|42.7|2.0|
>
> The results show that the proposed prompt format, which includes definitions and examples of each ambiguity type, performs similarly to the format used in the paper. **This indicates that the benchmark remains challenging and cannot be tackled through simple few-shot learning.** We will include this additional prompt format in the paper.
>
> > What is the style, task, and exemplar prompt?
>
> The few-shot prompt format used in the paper is provided in the attached PDF. Importantly, this prompt includes examples of both ambiguous questions with multiple correct SQL queries and unambiguous interpretations linked to a single query. These examples are sampled randomly with 5 seeds. We will expand the explanation in L563-564.
>
> >Could you provide the results of more advanced LLMs in few-shot settings since closed-source models still outperforms open-sourced LLaMa3-70B by a large margin?
>
> We were indeed surprised to find that LLaMa3-70B outperformed GPT-4o on our task, so we explored it further in the few-shot setting. However, we understand the reviewer’s concern and now provide 1-shot results for GPT-4o (with 3 seeds due to cost constraints):
> | Model|% Recall (Ambig)|% Recall (Unambig)|% Precision (Ambig)|% Precision (Unambig)|% AllFound|
> |---|---|---|---|---|---|
> |GPT-4o|32.7|58.4|47.6|45.1|4.1|
> |Llama3-70B|38.9|73.8|55.6|58.7|5.4|
>
> As the results show, LLaMa3-70B outperforms GPT-4o in the 1-shot setting as well. Note, there is no reason to expect GPT-4o to perform significantly better than with more examples (as the trend looks similar to Llama3-70B).
>
>
> **3. Concerns about biases**
> >use LLMs to generate could also introduce some bias in output distribution and cause unreality
>
> We manually verified the generated databases and did not observe any suspicious biases. As acknowledged in the limitations (L314-318), our databases may differ from production ones, but we believe they are sufficiently diverse for research and more realistic than in prior work.
>
> **4. Precision metric clarification**
>
> We measure precision as the number of correct queries divided by the total number of predictions.
>
> Ambiguous case:
> - If the model predicts one correct SQL query (one interpretation) or all correct queries (all interpretations), precision is 1.0 in both cases.
> - If incorrect queries are included, precision falls (e.g., 0.5 if half the predictions are correct).
>
> Unambiguous case:
> - Precision is 1.0 if the model predicts only the single gold SQL query.
> - If additional incorrect queries are included, precision is lower (1 / total predictions).
>
> In both cases, if there are no correct queries, the precision is 0. The similarity in precision for ambiguous and unambiguous questions may be due to models often outputting a single interpretation that happens to be correct (yielding 1.0 for both cases). The precision metric tends to weigh these cases more heavily than scenarios with multiple predictions, where only one is correct (resulting in a low score, e.g., 0.1).
>
> We will provide these clarifications in the paper.
>
> **5. Typos**
>
> Thank you for pointing out the typo! The table is correct, but our statistics in L187 and L206 are swapped.

---

> > ### Comment · Reviewer_zcDZ · 2024-08-19
> > **Solid Rebuttal**
> >
> > The authors resolve all my concerns, which include:
> >
> > 1. The foundation or basis of the ambiguity types. They listed all references to support their categories. Besides, different from previous work, this work focuses on the ambiguity in questions instead of database schema. I believe this is much more realistic and common in real life than pure morphological variations in controllable database schema items.
> >
> > 2. Few shot experimental results and with more advanced LLMs. Although the conclusions are somewhat counterintuitive, I choose to respect the experimental results cause the authors present detailed experiment settings, even with multiple random seeds. This proves that the proposed task cannot be tackled in a trivial way.
> >
> > 3. Regarding LLM generation biases, the authors claim that they have very carefully check the generated content. However, I will hold my view that pure LLM created data cannot completely replace human labor. A more feasible way should be at least contain some human labelled data (could be used as few-shot exemplar pools, maybe I missed somewhere), and use LLM empowered stages to simplify the annotation pipeline and include more data. However, this minor concern will not change my final attitude towards this work.
> >
> > 4. Precision metric. Clear and reasonable explanation. And think this deserves to be included in the paper cause this metric is not a common sense.
> >
> > Considering the extensive experimental results and authority sources of ambiguity definitions, which solve all my two major concerns, I am willing to raise my score to 7.

---

### Official Review · Reviewer_HeH4 · 2024-07-23
**valid problem, good execution, narrow coverage**

**Rating:** 7
**Confidence:** 5
**Correctness:** Everything is done properly.
**Clarity:** Yes.

**Review:**

The paper introduces a benchmark for text-to-SQL parsers by addressing the challenge of ambiguous user queries. Semantic parsing has significant practical applications, making advancements in this area impactful for various industries. The creation of a high-quality benchmark to drive research forward is both timely and valuable.

**Pros:**
1. **Practical Importance (S1)**: The topic of semantic parsing is crucial for real-world applications, and addressing ambiguity is a genuine problem in need of more robust solutions. A dedicated benchmark can drive significant advancements in this area.
2. **Quality and Clarity (S2)**: The paper is well-written and clearly outlines its objectives, methodology, and results. The execution of the idea is coherent and complete, suggesting that the final resource will be of high quality.
3. **Comprehensive Evaluation (S3)**: The paper benchmarks various advanced LLMs, demonstrating that the proposed dataset can serve as a valuable tool for measuring progress in handling ambiguities.

**Cons:**
1. **Limited Scope of Ambiguity (W1)**: The types of ambiguities considered (scope, attachment, and vagueness) are only a subset of the broader set of potential ambiguities. The authors should better define the scope of their work and position it in the context of the broader landscape of ambiguity issues in semantic parsing. For instance, the papers below discuss other facets of ambiguity that are not covered in this benchmark.

Zezhou Huang, Pavan Kalyan Damalapati, Eugene Wu:
Data Ambiguity Strikes Back: How Documentation Improves GPT's Text-to-SQL. CoRR abs/2310.18742 (2023) [TRL23 workshop]

Enzo Veltri, Gilbert Badaro, Mohammed Saeed, Paolo Papotti:
Data Ambiguity Profiling for the Generation of Training Examples. ICDE 2023: 450-463

Avrilia Floratou et al
NL2SQL is a solved problem... Not! CIDR 2024

While the benchmark’s focus on three types of ambiguity is a good start, expanding the types of ambiguities covered would make the dataset more comprehensive and valuable.

2. **Lack of patter variety in Data (W2)**: Given the construction process, the questions for scope and attachment ambiguities appear to be very similar, simply changing the entities/datasets involved. This can also explain the observed skew in predictions, with models tending to clearly favor certain patterns. This suggests that a large number of examples may not be necessary to capture these properties. Consequently, while the benchmark is large in size, it ultimately covers only a few types of ambiguity and patterns, limiting its overall impact.

3. **Realism and variety of Databases (W3)**: Again, given the construction process, the generated databases are validated for structure and content, but it is not possible to claim that they cover all kinds of data, especially if we think about proprietary, enterprise data, which can be have rare entities and concepts that are not well known to LLMs.
Minor, as also the author remark, ensuring that the generated more closely resemble real-world databases in complexity and naming conventions would further enhance the benchmark’s utility.

**Strengths:**

1. **Significance of Contribution**: Given the prevalence of ambiguities in real-world user queries, the benchmark is timely and relevant

2. **Relevance**: The benchmark is relevant to both academic researchers and industry practitioners.

3. **Quality of Research**: The submission is well-executed, providing clear and coherent methodology backed by experiments.

**Additional Feedback:**

n.a.

**Documentation:**

the supplementary material file contains a good datasheet that I believe is sufficient for the requirements of the track

**Ethics:**

no concerns

**Limitations:**

I found the limitations section clear about some of the limits of the proposed work.

**Opportunities For Improvement:**

**Scope of Ambiguity Types**: Expanding the range of ambiguities beyond scope, attachment, and vagueness would enhance the dataset's comprehensiveness and applicability.

**Clarification of Scope and Positioning**: Better defining the scope of the work and its position within the  landscape of ambiguity issues.

**Addressing Repetition in Data**: Reduce the similarity between questions for scope and attachment ambiguities to ensure that having more examples has a more meaningful impact on model evaluation.

**Relation To Prior Work:**

In general the paper is clearly positioned wrt to some of the work in the literature. However, as discussed above, there are several papers that are relevant but not covered.

**Summary And Contributions:**

The submission introduces a new benchmark for text-to-SQL parsers designed to handle ambiguous questions. The dataset contains 849 multi-table databases covering 16 domains and includes 4k+ SQL queries, covering three types of ambiguity: scope, attachment, and vagueness. The paper benchmarks several LLMs, revealing that even the most sophisticated models struggle to interpret ambiguity effectively, e.g., the best-performing model achieves ~30% recall on ambiguous questions. The benchmark aims to inspire the development of more capable semantic parsers and enable further research in handling ambiguous NL  queries.

---

> ### Author Rebuttal · Authors · 2024-08-16
>
> **1. Scope of ambiguity**
> >The authors should better define the scope of their work and position it in the context of the broader landscape of ambiguity issues in semantic parsing.
>
> Thank you for raising this important point. We provide a formal definition of ambiguity and clarified the scope of our work in the General Response to all reviewers. In summary, we define ambiguity as a **linguistic phenomenon** that arises from how questions are phrased, assuming the database context is known (both schema and values).
>
> **Discussion of Related Work:**
> - *Huang et al., 2023:* This work explores ambiguity in the KaggleDBQA dataset focusing on vagueness, underspecified output formats, and unknown data structures. In contrast, our work assumes the database context is fully specified, and we focus on different types of linguistic ambiguity in questions, not limited to vagueness.
>
> - *Veltri et al., 2023:* This work automatically generates declarative sentences containing facts that may lead to contradictions due to vague tables. Our approach, however, centers on human-written and verified questions that users might ask in real-world scenarios, rather than fact-checking. Additionally, our databases are multi-table and not only support vagueness but also include scope and attachment ambiguities.
>
> - *Floratou et al., 2024:* This study analyzes vague questions in KaggleDBQA and highlights issues in existing text-to-SQL benchmarks, where ambiguous questions are often linked to only one SQL query. AMBROSIA addresses this limitation by providing multiple SQL interpretations for ambiguous questions, while also covering a broader range of ambiguities beyond vagueness.
>
> In summary, this earlier work predominantly focuses on vagueness within small datasets (e.g., KaggleDBQA, ~100 vague questions) or relies on automatically generated data, which may lead to unrealistic cases. Our dataset, on the other hand, provides human-written interpretations and addresses scope and attachment ambiguities, which are often overlooked.
>
> We thank the reviewer for highlighting this relevant work. We will expand the related work section to incorporate these comparisons.
>
> **2. Patterns and data variety**
> >  Given the construction process, the questions for scope and attachment ambiguities appear to be very similar, simply changing the entities/datasets involved. This can also explain the observed skew in predictions, with models tending to clearly favor certain patterns.
>
> Evidence to the contrary is provided by our experimental results. If the data lacked variety and contained easily recognizable patterns, we would expect LLMs to learn these patterns from in-context examples. However, as shown in Section 4.3, even 5-shot examples (each including an ambiguous question, human-provided interpretations, and SQL queries) do not significantly improve performance. As noted in L296, the models’ tendency to favor one interpretation is consistent with the findings of Kamath et al. (2024). We argue that this reflects the inherent difficulty of capturing different interpretations, a challenge also identified by Liu et al., 2023, and Stengel-Eskin et al., 2024.
>
> We respectfully disagree that the questions for scope and attachment ambiguities are generated by “simply changing the entities/datasets involved.” All questions were written by human annotators and subsequently verified by the authors. Annotators were encouraged to vary and rewrite the provided templates, resulting in a diverse set of questions. We can confirm that the questions were written in various styles with linguistic variation. Furthermore, attachment ambiguity includes a range of sub-categories such as prepositional phrase attachment, ambiguities in relative clauses, and adjectives (L114-116), as well as four distinct configurations of databases (Figure 5). As demonstrated in recent robustness studies (e.g., Pi et al., 2022; Chang et al., 2023), existing text-to-SQL models are sensitive to such variations.
>
> Finally, to support our claims, we have also calculated the edit distance between the templates shown to annotators and their written questions and interpretations: **Scope: 9.2; Attachment: 12.3** (higher means the question is more different from the template; edit distance is 0 when there are no differences). This clearly demonstrates that annotators made substantial modifications to the templates. We will incorporate this analysis in the paper.
>
> Please note that our work is the first to introduce scope and attachment ambiguities, in addition to vagueness, in the context of text-to-SQL tasks.
>
> ---
> *Pi et al., 2022. Towards Robustness of Text-to-SQL Models Against Natural and Realistic Adversarial Table Perturbation. ACL 2022.*
>
> *Chang et al., 2023. Dr.Spider: A Diagnostic Evaluation Benchmark towards Text-to-SQL Robustness. ICLR 2023.*
>
> **3. Realism and variety of databases**
> >it is not possible to claim that they cover all kinds of data, especially if we think about proprietary, enterprise data, which can be have rare entities and concepts that are not well known to LLMs.
>
> We fully agree with the reviewer’s observation and explicitly state this as a limitation (L314-318). Our databases are designed to be “realistic” in terms of structure and content, avoiding the duplication issues seen in earlier work that relies on artificial processes for inducing ambiguity. Since our proposed benchmark yields low performance across models, we believe it is well-suited for academic research. Nevertheless, we hope that follow-on work will expand the dataset to further enhance its utility.
>
> We hope this response clarifies the scope of our work and the variety in our data. We are happy to answer further questions and kindly ask the reviewer to reconsider their score if these concerns have been resolved.

---

> > ### Comment · Reviewer_HeH4 · 2024-08-19
> >
> > thanks for the detailed comments
> > I will raise my rating to 7, please make sure to extend/clarify the positioning of your contribution wrt the existing efforts in the revised version

---

### Official Review · Reviewer_7sxp · 2024-07-24
**Review of Reviewer 7sxp**

**Rating:** 7
**Confidence:** 4
**Clarity:** The paper is straightforward and easy…

**Review:**

Overall, I enjoyed reading the paper with its sound observation of the phenomenon of ambiguity and the associated citations of previous works. The dataset construction process is reasonable and convincing. The only concern I have is the scope of the task being tackled in this work. I have elaborated on this aspect in the Opportunities For Improvement section.

**Strengths:**

- Well-written paper.
- High-quality dataset assessing text-to-SQL models' capability of handling ambiguous questions.
- Fresh perspective on the types of ambiguity in questions for text-to-SQL.
- The data generation pipeline is sound and reasonable.

**Additional Feedback:**

N/A

**Correctness:**

The motivation of the paper seems important, and although simplistic, the proposed approach of creating ambiguous questions is reasonable and convincing.

**Documentation:**

Yes.

**Limitations:**

The limitations of the paper are clearly stated.

**Opportunities For Improvement:**

Although I acknowledge the importance of this task and the authors have proposed a fresh perspective on ambiguity types, I am a little bit reluctant to give a higher score due to the scope of the task this paper tackles. Ambiguity is also prevalent in database values (or records) and, in fact, this can happen more often in real-world text-to-SQL use cases than the ones between questions and database schema (or having ambiguity at both the schema-level and value-level). As a result, this work seems to solve only part of the challenge in terms of resolving ambiguity in text-to-SQL.

**Relation To Prior Work:**

Yes.

**Summary And Contributions:**

This paper proposes a new dataset called Ambrosia, which includes different types of ambiguity, namely scope ambiguity, attachment ambiguity, and vagueness for text-to-SQL tasks. Through a novel approach of database generation and extensive human annotation of ambiguous questions, the proposed benchmark presents unique challenges that current state-of-the-art models, such as Llama3-70B and GPT models, find challenging. This leaves much room for improvement in existing text-to-SQL models for handling ambiguity.

---

> ### Author Rebuttal · Authors · 2024-08-16
>
> We thank the reviewer for their time and attention to our work.
>
> > Ambiguity is also prevalent in database values (or records) and, in fact, this can happen more often in real-world text-to-SQL use cases than the ones between questions and database schema (or having ambiguity at both the schema-level and value-level). As a result, this work seems to solve only part of the challenge in terms of resolving ambiguity in text-to-SQL.
>
>
> We would like to clarify that **AMBROSIA indeed covers cases where ambiguity arises at the value level**. For instance, databases with scope ambiguity in our dataset rely on many-to-many relationships between database *values* with a common element (Figure 1a). Furthermore, databases with attachment ambiguity are presented in 4 configurations: two with *schema-* and *value-level* ambiguity (Figures 5a and 5b) and two with *value-level* ambiguity only (Figures 5c and 5d).
>
> The only category with schema-level ambiguity and no value level ambiguity  is vagueness. Other text-to-SQL datasets that consider vagueness (Wang et al., 2023; Bhaskar et al., 2023) also do not focus on value-level vagueness. Moreover,  Wang et al. (2023) found in a study of real-world databases that **value-level vagueness is 4.5 times less common than schema-level vagueness (10% vs. 45%)**.
>
> We also provide a formal definition and a more detailed discussion of the scope of our work in the General Response to all reviewers.
>
> We will add this discussion into the paper. If our response has resolved your concerns, we would appreciate it if you could reconsider your score.

---

> ### Comment · Reviewer_7sxp · 2024-08-24
> **Official Comment by Reviewer 7sxp**
>
> Thank you for your kind response. I have improved my score.

---

### Official Review · Reviewer_1VY6 · 2024-07-26
**The paper presents a new benchmark for semantic parsing from tabular data. It addresses the issue of natural language query ambiguity.**

**Rating:** 9
**Confidence:** 3
**Correctness:** To the best of my understanding, yes …
**Clarity:** Yes the writing style is ok

**Review:**

The contribution is extremely relevant to the community. The problem of text-to -SQL becomes particularly complex while prompting unambiguous questions. I would prefer to see a more formal definition of ambiguity inspired by previous literature.
Apart from that, the contributions are quite convincing.

**Strengths:**

Novelty
Relevance to the field
Extensive experiments

**Additional Feedback:**

N/A

**Documentation:**

Yes

**Limitations:**

- The scope of the research is limited to the domains explored in the benchmark.

**Opportunities For Improvement:**

A better formalization of the concept of ambiguity is desirable.

**Relation To Prior Work:**

Yes, substantial differences with prior works

**Summary And Contributions:**

The contributions are theeefold. First, they release a new semantic parsing benchmark. Secondly they model three different types of ambiguity. Lastly, it explores the performance of various LLMs.

---

> ### Author Rebuttal · Authors · 2024-08-16
>
> Thank you for your feedback! We have provided a formal definition of ambiguity and a discussion of the scope of our work in the General Response to all reviewers. We will also incorporate this content into the extended version of our work.

---

> > ### Comment · Reviewer_1VY6 · 2024-08-20
> >
> > I appreciate the paper revisions, which address my concerns. I raised the overall score to 9.

---

### Author Rebuttal · Authors · 2024-08-16

We thank all the reviewers for their time and valuable feedback! We would like to provide the following clarifications:

**1. Formal definition of ambiguity and scope of the work**

We appreciate the feedback regarding the formal definition of ambiguity and have now included a more explicit definition inspired by Floratou et al., 2024:

**Definition 1:** Two SQL queries are **non-equivalent** if they produce different execution results, notwithstanding variations in layout or format.

**Definition 2:** Let $Q = {q_1, \dots, q_N }$ denote the universe of non-equivalent SQL queries that can be formulated given a database $D$, with known database schema and values. Let $s$ denote a NL question  and $f: s \rightarrow P(Q)$ a function that operates in the context of  database $D$ and deterministically maps $s$ to $P(Q)$, the power set of $Q$. We define $s$ as **ambiguous** if $f(s)$ has a cardinality of at least two.

In our work, we consider ambiguity as a **linguistic phenomenon**, which arises from the way a question is formulated, and leads to multiple  interpretations and corresponding SQL queries. This ambiguity persists because the database context does not uniquely resolve the  interpretations a question invites.

This definition excludes ambiguities related to data management issues (e.g., data format or coverage issues, the handling of NULL values), since we assume that the database schema and values are  known. Additionally, we do not consider underspecification of the output format (e.g., whether the result should contain only specific columns or if auxiliary columns are acceptable), as our prompts explicitly ask the model to output only the necessary columns.

While scope and attachment ambiguities are well-known examples of structural ambiguity, the research community has only recently started exploring them  in the context of LLMs (Liu et al., 2023; Kamath et al., 2024; Stengel-Eskin et al., 2024). Our work is the first to extend these concepts to the text-to-SQL task, broadening the scope of linguistic ambiguity beyond vagueness. We acknowledge that there are additional types of ambiguities beyond those covered here, and we hope follow-on work will expand on these.

**2. Data variety and task difficulty**

Regarding concerns about the variety and underlying patterns in our data: although the collection process for scope and attachment ambiguities used templates, all questions and interpretations were written and verified by human annotators (the edit distance between templates and responses ranges from 9 to 12). Furthermore, attachment ambiguity includes different sub-categories, such as prepositional phrase attachment, ambiguities in relative clauses, and adjective ambiguities (L114-116), as well as four different configurations of databases (Figure 5). The databases were designed to appear natural, but we do acknowledge (L314-318) that they might not fully resemble production databases.

Our experimental results further highlight that the dataset is challenging for LLMs, including GPT-4o. Even when few-shot examples are provided,  model performance does not improve, which in itself suggests that there are no simple patterns for the models to exploit.

We have added additional results for GPT-4o in the 1-shot setting, which proved to be the most effective for Llama3-70B:

| Model         | # Examples  | % Recall (Ambig) | % Recall (Unambig) | % Precision (Ambig) | % Precision (Unambig) | % AllFound |
| ------------- | ----------- | ----------------| ------------------ | ------------------- | --------------------- | ---------- |
| GPT-4o        | 1-shot      | 32.7            | 58.4               | 47.6                | 45.1                  | 4.1        |
| GPT-4o        | zero-shot   | 27.2            | 54.3               | 49.1                | 50.5                  | 0.4        |
| Llama3-70B    | 1-shot      | 38.9            | 73.8               | 55.6                | 58.7                  | 5.4        |
| Llama3-70B    | zero-shot   | 29.9            | 57.5               | 41.2                | 42.7                  | 2.0        |

We believe these additions and clarifications address the concerns raised and provide a better outline of our contributions.  We welcome any further discussion and hope our updates will assist the reviewers in their final evaluation.

---
*Floratou et al., 2024. NL2SQL is a solved problem... Not! CIDR 2024.*

*Liu et al., 2023. We’re Afraid Language Models Aren’t Modeling Ambiguity. EMNLP 2023.*

*Kamath et al., 2024. Scope Ambiguities in Large Language Models. TACL 2024.*

*Stengel-Eskin et al., 2024. Zero and Few-shot Semantic Parsing with Ambiguous Inputs. ICLR 2024.*

---

### Decision · Program_Chairs · 2024-09-26

**Decision:**

Accept (Spotlight)

**Comment:**

This is a paper that develops a new benchmark for semantic parsing for text-to-sql. The paper focuses on natural language ambiguity and identifies three different types of ambiguitity and studies how different LLMs behave for these settings.

The reviewers were very positive about this paper since it has a good dissection of ambiguity, is well executed and the authors did a good job in the rebuttals.